# Passive citizen science: The role of social media in wildlife observations

**Thomas Edwards**[1]*, **Christopher B. Jones**[1], **Sarah E. Perkins**[2], **Padraig Corcoran**[1]

**1** School of Computer Science and Informatics, Cardiff University, Cardiff, United Kingdom, **2** School of Biosciences, Cardiff University, Cardiff, United Kingdom

☉ These authors contributed equally to this work.
* EdwardsTJ1@cardiff.ac.uk

**Data Availability Statement:** All data files are available from the zenodo database (access from URL: https://zenodo.org/record/4527403#. YCL8Oi2I06g).

## Abstract

Citizen science plays an important role in observing the natural environment. While conventional citizen science consists of organized campaigns to observe a particular phenomenon or species there are also many ad hoc observations of the environment in social media. These data constitute a valuable resource for 'passive citizen science'—the use of social media that are unconnected to any particular citizen science program, but represent an untapped dataset of ecological value. We explore the value of passive citizen science, by evaluating species distributions using the photo sharing site Flickr. The data are evaluated relative to those submitted to the National Biodiversity Network (NBN) Atlas, the largest collection of species distribution data in the UK. Our study focuses on the 1500 best represented species on NBN, and common invasive species within UK, and compares the spatial and temporal distribution with NBN data. We also introduce an innovative image verification technique that uses the Google Cloud Vision API in combination with species taxonomic data to determine the likelihood that a mention of a species on Flickr represents a given species. The spatial and temporal analyses for our case studies suggest that the Flickr dataset best reflects the NBN dataset when considering a purely spatial distribution with no time constraints. The best represented species on Flickr in comparison to NBN are diurnal garden birds as around 70% of the Flickr posts for them are valid observations relative to the NBN. Passive citizen science could offer a rich source of observation data for certain taxonomic groups, and/or as a repository for dedicated projects. Our novel method of validating Flickr records is suited to verifying more extensive collections, including less well-known species, and when used in combination with citizen science projects could offer a platform for accurate identification of species and their location.

## Introduction

Observations on the distribution of wildlife species have always formed a crucial part of conservation and species management [1, 2], but are increasingly important in the face of rapid ecosystem changes that can be brought about, for example, by climate change and invasive

**Funding:** The authors received no specific funding for this work.

**Competing interests:** The authors have declared that no competing interests exist.

species, the consequences of which have implications for disease emergence and spread, as well as food security [2].

High-quality species distribution data are typically collected by professionals, but such data can be time-consuming and expensive to gather, and hence often lack broad coverage [1]. To overcome this knowledge gap, especially over a large spatial and/or temporal scale, citizen scientists are often engaged; members of the public who volunteer to record the presence of a given species and associated metadata, such as time, date, and location [2–4]. Thus, citizen science can be defined as the scientific work undertaken by members of the public, often in collaboration with or under the direction of professional scientists and scientific institutions, to collect data which in the context of this study relates to biodiversity [4, 5]. Citizen science projects can effectively crowdsource data, so amassing large volumes of species distribution data [4]. Due to the fact of using non-professionals, however, projects frequently come under criticism in terms of the accuracy of species identification, and associated data [6]. Further, organising citizen science campaigns and recruitment of volunteers can be a costly and challenging process [7]. More recent citizen science platforms have tried to address the problem of organising campaigns. An example of such a platform is iNaturalist (https://www.inaturalist.org/) a web-based and mobile-supported social network which allows individuals to upload photo observations and identify organisms [8]. However, the problems of attracting volunteers to participate, the correctness and open access of the uploaded data still remain.

Social network websites such as Flickr, Twitter, and Facebook have built a network of more than 2 billion users worldwide, generating millions of messages daily that are easily accessible, and reflect the observed reality of a quarter of the human population [9]. Social media websites have therefore emerged as an informal real-time information source that can contribute to the detection of trends and early warnings in critical fields such as ecological change, environmental problems, and shifts in ecosystems [9–11]. We define the use of social media that are unconnected to any particular citizen science program, but represent an unexploited source of valuable ecological data as passive citizen science. In contrast to citizen science campaigns, the passive citizen science approach provides a cost and time-efficient method for collecting wildlife-related data on a larger scale and for a wider time-span. Similar to the citizen science approach, it involves the participation of non-experts. However, datasets are collected without the organisation and conduct of specific campaigns or restrictions on species observations and time-frames. Instead it consists of a process of crowdsourcing in which data are retrieved from Internet resources, particularly social media, to which members of the public have uploaded observations such as annotated photos of wildlife.

A quantitative review of the application of social media in environmental research, conducted by [12] suggests a very rapid growth in the field of environmental monitoring, with Twitter and Flickr being most frequently used as data sources. Among the identified strengths of social media are the large volume of available data samples which makes data collection a less labour-intensive, time-consuming and costly procedure [12–14]. Social media data allows for a timely and (near) real-time monitoring and analysis of species distribution ([12, 15, 16]).

Despite the potential of social media to be used for species distribution models there are still some concerns about the quality and reliability of information mined from social media [9, 12, 17]. There are also concerns about the data ownership and future availability of social network data [9, 12, 18].

Arguably, the geotags given to photos on social media sites can be more reliable than user-submitted data as they are assigned automatically by GPS location systems, and if automatic identification of species can be employed such an approach has the potential to outweigh the skills of the general populace. It is for these reasons that the photo sharing site, Flickr, has been

recognised as a particularly valuable resource in ecology that could contribute to species distribution models [2, 9, 19].

The use of internet sources for gathering wildlife-related data in citizen science initiatives has emerged in recent years [20–22]. An example includes urban residents reporting occurrences of tagged birds through a Facebook group, a smartphone application and email [20]. A crowdsourcing tool was employed in [21] to collect data for the creation of a land cover map, while in [22] crowdsourcing is used as a supplemental method for collecting hydrologic data. An overview of the impact of internet social networks on traditional biodiversity data collection methods in [10] is optimistic and concludes that social media can potentially play an important role in conservation science. Additionally, the authors of [23] present a review of the iEcology approach which encompasses the use of automated tools for discovering patterns in the natural world using data accumulated in digital sources collected for other purposes. The authors highlight the value of social media such as Flickr. The use of iEcology approaches is increasing as it provides low-cost and fast data collection, pattern identification, and visualisation of nature-related data. In particular, the study of [11] investigates whether a plant species image classifier can be used to extract relevant plant observations from Flickr, using a general search term of 'flower'. Analyses showed that automated methods have the potential to help identify wildlife-related imagery data on social media, especially when photos were focused on single native species in rural situations or when classification was performed at genus or class level. It was suggested that future work could usefully focus on searching for individual species including invasive species. Our work does indeed do that and differs from [11] in providing a systematic evaluation of the detection of a large number of different types of wildlife species (not just plants) relative to existing citizen science data. Other recent research by [24, 25] focused on investigating how automated methods for identifying species from imagery data compare to manual annotations performed by citizen scientists. Results showed that there are still challenges in using automated approaches for verifying wildlife-related imagery such as imbalanced datasets and visual similarities between species. However, the low-cost and potential efficiency of such methods are the motivation for continuing research in this area.

Here, we focus specifically on the potential of Flickr for collecting species distribution data as it hosts one of the most extensive and easily accessible collections of geo-referenced photos of its kind and, because it is photo-based, it enables the possibility to validate observations by comparing the asserted species name, as provided in a tag or caption, with the content of the image. We assess the value of species distribution data gathered from the Flickr photo-sharing website relative to existing content on a public source of biodiversity data, the UK National Biodiversity Network portal. The National Biodiversity Network (NBN) Atlas (https://nbn.org.uk) portal holds the most extensive collection of biodiversity information within the UK with over 220 million species occurrences. It is a charity, with a membership including many UK wildlife conservation organisations, government, countryside agencies, environmental agencies, local ecological records centres and many voluntary groups dedicated to citizen science. It has more than 200 members and allows recording, sharing, displaying, and downloading species records. Records of species sightings are uploaded to the Atlas by registered organisations where datasets are collected by citizen science campaigns [26]. NBN datasets have previously proved useful in studying distribution patterns of UK species [26, 27].

Our aim is to explore the potential of social media to supplement species distribution data, and in doing so to serve as a form of passive citizen science. We conducted analyses with two case studies, one being the 1500 species that were most frequently recorded on NBN and the other being invasive species in the UK that have records on NBN. In comparing species distributions from Flickr with those of the NBN we quantify the value of social media acquired

distribution data on the largest number of species considered to date in such studies. Our approach uses a novel method of validating Flickr species images with the Google Cloud Vision API, that extends the method presented in [16] by automatic matching of the assigned categories to the content of a hierarchical species taxonomy.

Research analogous to our own has been carried out previously, but on a much smaller scale, and in a time-consuming manner [2, 9, 16]. In the forementioned research, the authors evaluate social network sites (Flickr and Twitter) relative to biodiversity data portals in order to identify the potential use of ad-hoc methods for augmenting traditional citizen science data collections. This previous research was conducted on a narrow range of species (between two and four). Another similar research by [11] investigate whether an image classifier for identifying plants could facilitate the discovery of unexploited biodiversity data from Flickr. However, this approach is focused purely on species occurrence on Flickr and thus does not provide a clear evaluation of the role of social sites observations compared to more traditional approaches.

Our validation approach is similar to that used in [16] to verify Flickr data. There the Google Reverse Image Search was used to return labels that best describe the content of a given photo. All labels per species were ordered in descending order of frequency. Then, the authors manually identify which are the species-relevant and irrelevant tags, among the most frequent ones, that can help indicate which photos are true representations of the given species. Despite the benefits of such an approach, especially when photos need to be evaluated only for a couple of species, it is unsuitable for the validation of larger collections where the manual upload of photos and manual selection of relevant tags per species can be a time-consuming process. We propose a fully automated image verification approach suitable for verifying large and diverse species collections. Specifically, we deploy the Google Cloud Vision API which allows fully automatic image verification. Further, we reduce the incidence of missed matches by employing a species taxonomy that supports matching between alternative names for a species as well as generic matches between terms in the relevant species hierarchy that were not used in the Flickr tags.

In summary, there are several limitations of previous research on using social media to augment traditional biodiversity portals, in that the analyses have been performed on very small numbers of species, the methods for accessing the social media are either manual or only partly automated, and the results are limited in the degree of verification.

## Materials and methods

We perform three types of analysis to compare species occurrence between the NBN and Flickr, consisting of a summary statistical analysis and spatial and temporal analyses. The statistical analysis compares the frequency of occurrence of species between the two data collections, performed on different taxonomic levels of species and class. The spatial analysis determines whether Flickr species observations match by location the NBN species observations. Because many species have variable distributions and abundances throughout the year we also use a temporal analysis to compare the time patterns of the NBN and Flickr data collections. We compared the locations of data occurrences for the two data sources for a time span of 3 months, 6 months and 12 months. We verify Flickr species identification through an image content verification approach using the Google Cloud Vision API to identify objects that appear in a given photo. The Google Cloud Vision API labels images with multiple taxonomic categories (i.e. labels) ranging from general to specific. Our image validation approach is based on coarse matching between all species names following down from the class of a species and the labels returned by Google Cloud API. In this way, we avoid a potentially high

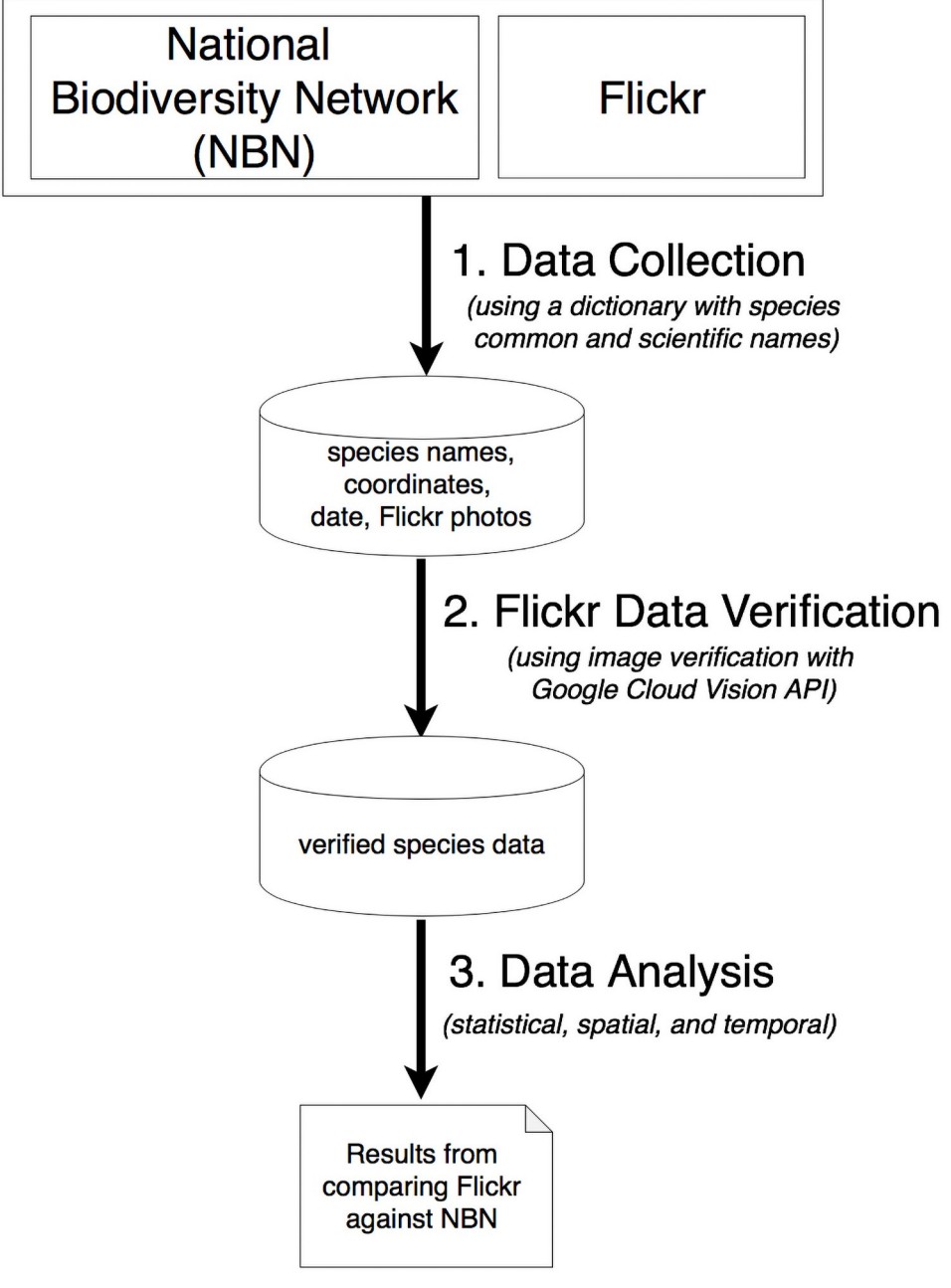

**Fig 1. Methodology overview.**

number of false negatives for less common species that are less likely to be identified on the API at the species level but might be identified at higher taxonomic levels. An outline of the methodology is depicted in Fig 1 and each step is detailed below.

## Data collection

**NBN data collection.** The NBN was selected as the biodiversity data portal for our study because it holds the most extensive collection of biodiversity information within the UK. We collected the names and the number of occurrences for the top 1500 species on NBN using the

NBN Atlas Occurrence Facet Search. We performed our search over all collections within the NBN and limited it for the territory of the UK.

For each of the species retrieved from the NBN we obtained, via a search on the NBN, all the alternative names associated with the species (scientific name and common names), the NBN species ID, and its taxonomic classification hierarchy. The names associated with each of the species were used for downloading data from Flickr. The taxonomic classification hierarchy is used for the verification of the Flickr data collection in combination with the Google Cloud Vision API. The NBN service does not support exact match between a given search term and the species name given to a record. Instead, the NBN service does partial matching between the search term and the species record names. For instance, the search term 'brown squirrel' can return results for other types of squirrels such as red squirrels, because there is a partial match between the search term *'brown squirrel'* and the record name *'red squirrel'*, i.e, the word 'squirrel'. Therefore, downloaded records can sometimes include species which are irrelevant to the search term. Consequently, we remove the species records irrelevant to the search term.

Further to that, some records are incomplete, lacking temporal or geo-information. To address this we filtered out irrelevant records and those with missing information. For inclusion in our dataset each record constituted record ID, geo-coordinates of the occurrence, date of the occurrence, NBN species ID.

**Flickr data collection.** Using the Flickr API interface we used both the scientific and common names, and limited our search to geo-tagged posts within the UK. Our search was therefore based on downloading posts with tags matching at least one of the alternative names given for a species in NBN. We downloaded the following types of information from Flickr: image coordinates, 'taken date' and 'posted date' of the post, post id, the image associated with the post, title, and all the tags associated with the post. 'Taken date' refers to the time at which the photo was taken while 'posted date' represents the time at which the photo was uploaded to Flickr. For performing the temporal analysis, we use the 'taken date'. However, early observations showed that 'posted date' and 'taken date' do not differ more than 3 months for the majority of Flickr posts. Additionally, we consider only posts where coordinates are extracted from a GPS-enabled device which is either the device used to upload the photo, or the device used to take the photo. We ignore posts associated only with user-provided locations.

## Flickr data validation

For describing the image validation method we heavily rely on types of information, clarified in Table 1. Flickr images needed to be validated because the content of the photos uploaded with associated tags might not match the species name tags given by the Flickr users. However, existing image verification approaches lack the ability to scale to large collections of species and they often need manual or semi-automatic verification. Further, a common limitation of automated image verification methods is the inability to accurately distinguish between species with similar visual characteristics. Therefore, performing a match between species names and

**Table 1. Main concepts used to describe the image verification method.**

| Concept | Description |
|---|---|
| Flickr Tags | The tags Flickr users have given to the photos they have uploaded on Flickr |
| Google Cloud Vision API labels | These are the labels returned by the image recognition API for the objects recognized in each Flickr photo. We use the Google Cloud Vision API to verify whether the species on the Flickr photos represent the given species |
| NBN species names | The list of species names extracted from NBN species classification taxonomy |

image recognition system labels would result in many photos being regarded as invalid representations of the species, which limits the coverage of methods. We aim to address this issue and provide an approach for verifying large and diverse image-related species data fully automatically by using a Bag-of-Words (BOW) approach. Specifically, we use Google Cloud Vision API to coarse match between all names following down from NBN species taxonomic class and the labels returned by Google Cloud Vision API. A potential problem of the BOW approach is that on lower levels of the taxonomy it might not be able to distinguish between species belonging to the same class such as two different types of grass. However, we hypothesise that for diverse species collections such a coarse match-based approach will help improve the coverage of automated image verification methods (recall measure) without affecting their accuracy and precision. Further, we compare the BOW approach against a simple 'exact match' approach in which we confirm a match if the tags returned by the Google Cloud Vision API include the species name. This baseline method could be regarded as analogous to (but differing from) that of [16] which also considers the tags generated by an image recognition system. In that case however, the process of identifying whether images are of a particular species consists in filtering out image matches for which the the most frequent generated labels are deemed irrelevant. Further, we expand our analysis by performing evaluation at the genus-level. In this way, we compare three approaches for evaluating large collections of species-related image data within large collections of species, i.e. exact match, class-level match, and genus-level match.

The main steps involved in our image verification approach are illustrated in Fig 2. The methodology consists of the following steps: First, for each Flickr image, we download all species names from NBN following down from the species taxonomic class (for class-level match) or genus (for genus-level match). We perform image verification using Google Cloud Vision API model and store the labels returned by the model. Then, we apply a coarse match between the NBN species names and the Google Cloud Vision API labels. If there is a match between the labels returned by the image verification model and the NBN species-related names, then the image is considered a correct representation of the species given by the Flickr user.

Google Cloud Vision API is however not trained on wildlife data and thus some of the less well-known species names might not be returned as labels, for instance, 10-spot Ladybird (*Adalia decempunctata*) and 22-spot Ladybird (*Psyllobora vigintiduopunctata*). Also, species belonging to the same class (e.g. 'cuckoo' and 'sparrowhawk') might have very similar visual appearance and thus the Google Cloud Vision API cannot be assumed to distinguish between the two species. Therefore, using exact matching between species names and Google labels will lead to a high number of false negatives.

An example of a Google Cloud Vision API result for a single photo correctly tagged on Flickr as Adder, gives the following categories: *Reptile (98%), Snake (98%), Scaled reptile (93%), Viper (91%), Serpent (89%), Terrestrial Animal (87%), Rattle Snake (84%), Sidewinder (70%), Adaptation (67%), Colubridae (65%), Eastern Diamondback Rattlesnake (56%), Elapidae (53%)*. The higher the score, the more confident the assignment of the category is for the given image, where the score is given in brackets next to the tags.

The photo labels returned by Google Cloud Vision API can be organised as a taxonomy that matches the species taxonomy returned by NBN.

Fig 3 displays the NBN classification for Adder and the labels returned by Google Cloud Vision API for this photo. We use the NBN taxonomic classification for the species to choose relevant species names to match the labels returned by Google Cloud Vision API. A BOW approach is adopted where we treat the names in the classification hierarchy for a species as a list of names ignoring the hierarchical and semantic relations between these names. We consider all names in the classification hierarchy following down from class, and we match these

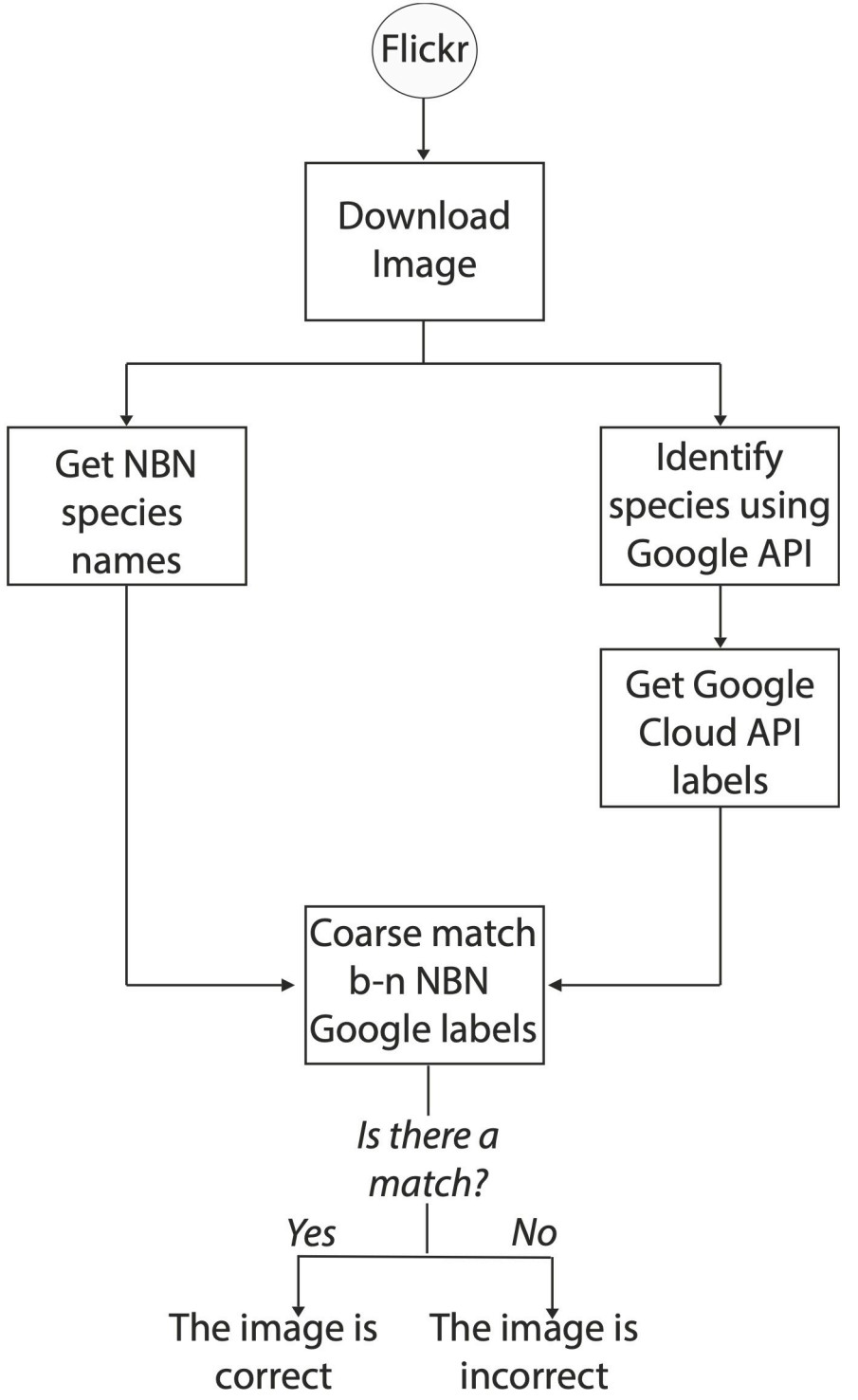

**Fig 2. Image validation approach overview—'b-n' refers to between.**

terms to the labels given by Google Cloud API. In the example, given in Fig 3, the use of the classification finds an exact match between the species name 'Viper' (an alternative name for 'Adder') and the Google Cloud Vision API term Viper. Using exact matching on the Flickr label of Adder would not have found any match, resulting in a false negative for this

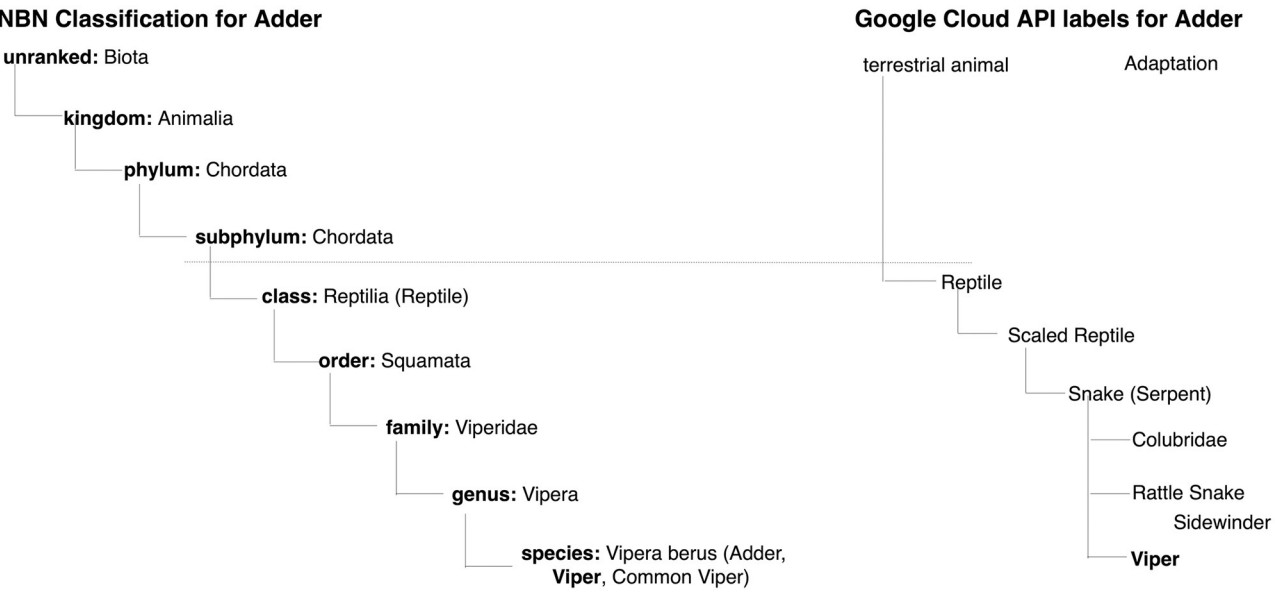

**Fig 3. Google Cloud Vision API label taxonomy and NBN classification for Adder.**

observation. Another example is for species *Phleum pratense (Timothy Grass)*, which is from class Magnoliopsida and family *Poacae (Grass)*. Google Cloud Vision API returns for images with this species the label 'grass', rather than 'timothy grass' and thus coarse match would be successful in this case.

Note that we use both scientific and common names for matching, as both can occur within the NBN derived taxonomy and the labels returned by the Google Cloud Vision API. We performed manual verification of the Google results for 50 randomly selected species where we randomly select 40 images per species. We used these 2000 images to evaluate the performance of our image verification method.

## Data analysis

The data analyses are based on two case studies: the 1500 most frequently recorded species on NBN and the invasive species in the UK that appear in both data collections. Spatial comparison between the NBN and Flickr datasets was performed using spatial grid modelling, in which geographic space is divided into regular grid cells. The cells were classified according to whether they contained observations from one or other or both of the two sources. The classification was further refined according to time windows to support both a spatial and a temporal analysis. By varying grid cell sizes, and cell aggregation (i.e. one by one vs three by three), as well as the time window, we performed a number of scale-variant spatio-temporal analyses.

There were two main methods of performing spatial analysis:

1. One by one cell comparison: We compare Flickr and NBN species occurrence data per 10km, 20km, and 40km grid square. We calculate a confusion matrix, which is used to describe the performance of a classifier on a test data set for which the true values are known, where Flickr is the test data set and NBN the true values. The cells of the confusion matrix are defined as follows:

   - 'True Positive' (TP): a cell has both NBN and Flickr data points for the species

   - 'True Negative' (TN): a cell does not have occurrences from either of the sources

- 'False Negative' (FN): a cell has no Flickr data for the species, but it does have NBN data for the species

- 'False Positive' (FP): a cell has Flickr data for the species but no NBN data

2. Three by three cell comparison: We compare Flickr and NBN using a three by three analysis centred on every cell. In this approach, we count a true positive if there is a Flickr posting in a cell and if there are NBN records within either the cell itself or in any of the adjacent eight cells. A false negative would be declared if a set of nine cells had at least one NBN record but no Flickr record. A false positive indicates if there is a Flickr posting in a cell, but there are no NBN records within either the cell itself or in any of the adjacent eight cells. A 'True Negative' would be no Flickr postings and no NBN records in any of the nine cells.

Based on the measures above we compute precision, recall, and F1-measure. Recall is calculated by dividing the number of True Positives by the True Positives plus the False Negatives ($\frac{TP}{TP+FN}$). Thus if there were 10 cells containing NBN data, and for each of them Flickr data were also present, then the recall would be 100% or 1.0.

Precision is calculated by dividing the number of True Positives by the False Positives plus the True Positives ($\frac{TP}{TP+FP}$). In the previous example if, in addition to the 10 cells containing both NBN and Flickr data, there were a further 5 cells that contained Flickr data but no NBN data, then the precision would be 66% or 0.66.

F1 Score is calculated using recall and precision. It is used because precision and recall alone are not an accurate representation of one data set's superiority over another, as one could have better precision and the other a better recall. The F1 Score provides a harmonic mean that gives a clearer view of a dataset's accuracy when compared with ground truth. It is computed as double the product of precision and recall divided by the sum of precision and recall($\frac{2*precision*recall}{precision+recall}$).

We look at temporal accuracy of Flickr on seasonal (3 months), half yearly (6 months) and yearly patterns (12 months). The seasonal (3 month window) and half-yearly patterns (6 month window) aim to reveal movement patterns affected by climate and environmental changes typical for the different seasons and times of the year. This allows us to identify seasonal patterns that are usually unaffected by yearly changes such as seasonal migrations. For these purposes, we ignore the year for the seasonal and half yearly analysis and instead we perform calculations across all years.

The total period for the temporal analysis is between 2004–2018 (2004 is the year in which Flickr was launched and 2018 is the year when we finalised the Flickr collection process). For the seasonal analysis (3 months) we consider time windows reflecting the four seasons with no overlaps between the months (spring (March, April, May), summer (June, July, August), autumn (September, October, November) and winter (December, January, February)). The half yearly analysis are performed considering two windows, one ranging from September to February and the other encompassing the warmer seasons from March to August.

**Ethics statement.**   The use of the datasets was done in compliance to the National Biodiversity Network and Flickr Terms and Conditions and PLOS ONE requirements for this type of study.

## Results and discussion

### Statistical analysis

**NBN and Flickr datasets comparison.**   Across the 1500 most numerous species on NBN Atlas, 90% were found on Flickr and 100% of species in the Flickr dataset were found on NBN

**Table 2. The top 10 species on Flickr with the highest number of records.**

| Scientific name | Common name | Flickr count | NBN count |
|---|---|---|---|
| *Hyacinthoides non-scripta* | *Bluebell* | 20,940 | 54,893 |
| *Bellis perennis* | *Daisy* | 20,656 | 28,748 |
| *Erithacus rubecula* | *Continental Robin* | 19,248 | 3,938,616 |
| *Morus bassanus* | *Gannet* | 17985 | 14252 |
| *Fagus sylvatica* | *Beech* | 15,842 | 24,973 |
| *Hedera helix* | *Ivy* | 14,474 | 27,211 |
| *Anas platyrhynchos* | *Mallard* | 13,500 | 834,039 |
| *Taraxacum officinale agg.* | *Dandelion* | 13,443 | 27,269 |
| *Pteridium aquilinum* | *Bracken* | 12,708 | 30,741 |
| *Phleum pratense* | *Timothy Grass* | 9,000 | 11,903 |

Atlas. The NBN Atlas records, as expected, far outnumber those on Flickr, being 93,656,179 and 791,059 respectively. It is worth noting that NBN data used here covers the entire collection period; 1800-2018 while Flickr data covers only 2004-2018. It was found that 35% of the species counted on Flickr have more than 100 occurrences. Table 2 lists the top 10 most frequently recorded species on Flickr (mostly with more than 10000 occurrences).

The best represented species on Flickr (see Table 2, can be split into three main categories: pretty, ie. photogenic, flowers (Bluebell, Daisy, Dandelion), sessile green plant species (Ivy, Beech, Bracken) and garden and aquatic birds, which are also diurnal (Continental Robin, Mallard). Notably all are easily accessible. These same patterns were mirrored at the class level with the highest number of returns for Flickr being *Magnoliopsida*, a class of flowering plants, and the second highest was *Aves*. These results also support findings presented in [11] where results showed that Flickr is rich in plant-related data mainly dominated by non-native flowering plants. However, that research focused only on plants and does not perform comparison between Flickr and citizen science data portals. *Phleum pratense (Timothy Grass)* as a well documented species in Flickr is an interesting observation as, compared to the other commonly observed species (see Table 2), it is not a well known species that is readily identified, suggesting that it was incidental in many images and Flickr may be good at picking up species that appear as a background in a photo. Another example of such a species is *Hedera helix (Ivy)*.

NBN and Flickr datasets are similar in the diversity of classes they represent with the ten best represented classes in both collections being the same, with the same three most common classes of *Insecta* (Insects), *Magnoliopsida* (plants), and *Aves* (birds). Both data collections are representing species from a small number of classes. This suggests that the same observer bias in photos also occurs in NBN data collections.

The top 10 species on NBN are garden birds (see Table 3), and they are represented well in the Flickr dataset with occurrences in most cases above a thousand.

**NBN and Flickr datasets comparison for invasive species in the UK.** There are 82 invasive species for UK that also have occurrence records on NBN. The total count of records of invasive species on NBN is 1,485,744. The total number of Flickr posts for the invasive species that are also recorded on NBN is 27,187. The number of species with occurrences above 100 for both NBN and Flickr data collections is 19 (of 82), which is 23% of the number of invasive species on NBN.

The invasive species with more than 100 occurrences for both NBN and Flickr are diurnal mammals, birds (more than 50%) along with a few "pretty" flower species (see Table 4).

The best represented invasive species on Flickr are *Branta canadensis (Canada Goose)*, *Scirurus carolinensis (Grey Squirrel)*, *Gallinago gallinago (Snipe)*, *Oryctolagus cuniculus (Rabbit)*,

**Table 3. The top 10 species on NBN with the highest number of records.**

| Scientific name | Common name | NBN count | Flickr count |
|---|---|---|---|
| *Turdus merula* | *Blackbird* | 4,609,821 | 3,234 |
| *Cyanistes caeruleus* | *Blue Tit* | 4,164,338 | 3,491 |
| *Erithacus rubecula* | *Continental Robin* | 3,938,616 | 19,248 |
| *Columba palumbus* | *Woodpigeon* | 3,584,436 | 1,660 |
| *Prunella modularis* | *Dunnock* | 3,513,651 | 2,179 |
| *Parus major* | *Great Tit* | 3,507,350 | 2,670 |
| *Fringilla coelebs* | *Chaffinch* | 3,444,776 | 3,474 |
| *Passer domesticus* | *House Sparrow* | 3,184,175 | 2,312 |
| *Streptopelia decaocto* | *Collared Dove* | 3,094,475 | 929 |
| *Chloris chloris* | *Greenfinch* | 2,900,214 | 2,030 |

*Rhododenron ponticum (Rhododendron)*, *Aix galericulata (Mandarin Duck)*, and *Cygnus atratus (Black Swan)*. The species for which NBN and Flickr have a similar number of records are *Sus scrofa (Wild boar)* and *Bubo bubo (Eurasian Eagle Owl)*.

## Flickr data verification

In initial exploratory work, we performed tests with the tags returned by the Google Cloud Vision API. We found that the tags with a score above 60% are more likely to imply the correct species displayed on the photos. The tags with a score lower than 60% usually describe either less relevant objects of the photo, e.g. parts of the background ('*leaf*'), characteristics of the animal ('fawn'), or are names of species that are irrelevant to the photo ('Diamondback Rattlesnake' when the species is Adder). Therefore, we used only tags with a score higher than 60%.

**Table 4. Species occurrences for NBN and Flickr for invasive species with number of occurrences above 100.**

| Scientific name | Common name | NBN species count | Flickr species count |
|---|---|---|---|
| *Branta canadensis* | *Canada Goose* | 377,111 | 3,328 |
| *Sciurus carolinensis* | *Grey squirrel* | 350,113 | 3,249 |
| *Gallinago gallinago* | *Snipe* | 325,210 | 1,619 |
| *Oryctolagus cuniculus* | *Rabbit* | 96,093 | 7,994 |
| *Alopochen aegyptiacus* | *Egyptian Goose* | 31,591 | 862 |
| *Rhododenron ponticum* | *Rhododendron* | 30,803 | 3,489 |
| *Branta leucopsis* | *Barnacle Goose* | 24,269 | 289 |
| *Aix galericulata* | *Mandarin Duck* | 19,693 | 1,500 |
| *Muntiacus reevesi* | *Reeve's muntjac* | 16,428 | 489 |
| *Cygnus atratus* | *Black Swan* | 8,761 | 1,148 |
| *Buddleja davidii* | *Buddleia* | 5,654 | 443 |
| *Heracleum mantegazzianum* | *Giant Hogweed* | 5,348 | 190 |
| *Anser caerulescens* | *Snow Goose* | 5,085 | 177 |
| *Anser indicus* | *Bar-Headed Goose* | 3475 | 164 |
| *Aix sponsa* | *Wood Duck* | 2,688 | 290 |
| *Cervus nippon* | *Sika Deer* | 2,442 | 226 |
| *Chrysolophus pictus* | *Golden Pheasant* | 1,745 | 167 |
| *Sus scrofa* | *Wild boar* | 441 | 373 |
| *Bubo bubo* | *Eurasian Eagle Owl* | 395 | 537 |

In the rest of the section, we evaluate our image verification approach (class level BOW approach) against two other image verification approaches. The first one is a fully automated 'exact match' approach, as explained earlier in the section on Flickr Data Validation, where we perform an exact match between the NBN species names associated with a given species and the Flickr posts names. The second approach is a genus level BOW approach where we consider names following down from the genus of the species. Evaluating approaches considering different levels of coarse match (species-level, genus-level, and class -level) allows us to judge which approach is more suitable for evaluating diverse collections of species without affecting the precision.

The average results, presented in Table 5, show that class-level BOW approach outperforms the other two evaluation approaches by a significant margin with F1 = 0.79 versus F1 (baseline) = 0.20 and F1 (BOW (genus level)) = 0.27. Specifically, the class-level BOW method achieves best results, compared to the other approaches, for 45 out of the 50 species. For 14 of these species, the class-level BOW method has equal or slightly lower precision than the other two methods, however the recall is much higher in these cases leading to a better performance overall. For instance, for 'Passer domesticus (House Sparrow)', the baseline and BOW (genus level) approach have a higher precision of 1.0 while BOW (class level) has a precision of 0.97, however the recall (0.85) is more than double compared to the recall of the other approaches. This indicates that class-level BOW method is a more suitable for evaluating wider range of species than the other two approaches without significantly affecting the precision.

It should be noted however that the presence of some species for which none of the approaches was successful, i.e., 'Erica cinerea (Bell Heather)', 'Stachys officinalis (Betony)', 'Solanum dulcamara (Bittersweet)', 'Hyacinthoides non-scripta (Bluebell)', and 'Acer campestre (Field Maple)', shows that there are species for which the BOW approach is unsuitable. Thus other methods need to be considered that might include species characteristics (color, shape), attributes (feather, beak, etc.). It might also be a reflection of the lack of effectiveness of the image classifier.

The most common causes of false positives for BOW are photos that include an artificial representation of a species, such as a boat with a figure of a goose (Fig 4), and hence do not represent a living species. Common cases of false negatives for BOW are photos which include the species but the focus of display is another object. In the example given in Fig 4, the main object in the photo is a building, and thus Google Cloud Vision API returns labels associated with the building and the characteristics of the building, rather than the plant (i.e. Hedera helix (Ivy)). Similar issues were highlighted for example in [11, 16].

## Spatial and temporal analysis

**The top 1500 species on NBN.**   The average precision and recall across all species for each type of spatial and temporal constraint for one by one grid cell analysis is 0.38 (38%) for precision and 0.2 (20%) for recall. The recall score shows that on average 20% of all NBN data was also reflected by the Flickr data. The precision score shows that on average 38% of the Flickr cell-based identifications of a species were also reflected in the NBN data Fig 5.

The average precision and recall across all species for each type of spatial and temporal constraint for three by three analysis is 0.6 (60%) for precision and 0.1 (10%) for recall (see Fig 5). In comparison to one by one analysis, the average precision for three by three analysis is higher ranging from 0.27 (27%) to 0.78 (78%) for the different cell sizes while recall tends to be lower and does not vary much for the different cell sizes. The number of false negatives is significantly higher for three by three analysis and thus the recall value is lower. The reason for this is can be attributed to the wider range of species recorded within the NBN in comparison to the

**Table 5. Comparison between class-level and genus-level BOW image verification approaches and the baseline approach, based on exact-match approach.**

| species | baseline | | | BOW (genus level) | | | BOW (class level) | | |
|---|---|---|---|---|---|---|---|---|---|
| | p | r | F1 | p | r | F1 | p | r | F1 |
| Coccinella septempunctata (7-spot Ladybird) | 0.00 | 0.00 | 0.00 | 0.00 | 0.00 | 0.00 | **0.97** | **0.95** | **0.96** |
| Propylea quattuordecimpunctata (14-spot Ladybird) | 0.00 | 0.00 | 0.00 | 0.00 | 0.00 | 0.00 | **1.00** | **1.00** | **1.00** |
| Vipera berus (Adder) | **1.00** | 0.65 | 0.79 | **1.00** | 0.65 | 0.79 | **1.00** | **0.76** | **0.87** |
| Tyto alba (Barn Owl) | 0.00 | 0.00 | 0.00 | 0.00 | 0.00 | 0.00 | **0.89** | **0.50** | **0.64** |
| Ophrys apifera (Bee Orchid) | 0.00 | 0.00 | 0.00 | **1.00** | 0.82 | **0.90** | **1.00** | **0.83** | **0.90** |
| Erica cinerea (Bell Heather) | 0.00 | 0.00 | 0.00 | 0.00 | 0.00 | 0.00 | 0.00 | 0.00 | 0.00 |
| Stachys officinalis (Betony) | 0.00 | 0.00 | 0.00 | 0.00 | 0.00 | 0.00 | 0.00 | 0.00 | 0.00 |
| Solanum dulcamara (Bittersweet) | 0.00 | 0.00 | 0.00 | 0.00 | 0.00 | 0.00 | 0.00 | 0.00 | 0.00 |
| Turdus merula (Blackbird) | 0.96 | 0.76 | 0.85 | 0.95 | 0.75 | 0.84 | **0.91** | **1.00** | **0.95** |
| Hygrocybe conica (Blackening Waxcap) | 0.00 | 0.00 | 0.00 | 0.00 | 0.00 | 0.00 | **1.00** | **0.60** | **0.75** |
| Cyanistes caeruleus (Blue Tit) | 0.00 | 0.00 | 0.00 | 0.00 | 0.00 | 0.00 | **0.97** | **0.89** | **0.93** |
| Hyacinthoides non-scripta (Bluebell) | 0.00 | 0.00 | 0.00 | 0.00 | 0.00 | 0.00 | 0.00 | 0.00 | 0.00 |
| Buteo buteo (Buzzard) | **1.00** | 0.52 | 0.68 | **1.00** | 0.52 | 0.68 | **1.00** | **0.81** | **0.89** |
| Corvus corone (Carrion Crow) | 0.00 | 0.00 | 0.00 | 0.00 | 0.00 | 0.00 | **1.00** | **0.95** | **0.97** |
| Fringilla coelebs (Chaffinch) | 0.00 | 0.00 | 0.00 | 0.00 | 0.00 | 0.00 | **1.00** | **0.89** | **0.95** |
| Periparus ater (Coal Tit) | 0.00 | 0.00 | 0.00 | 0.00 | 0.00 | 0.00 | **0.97** | **0.92** | **0.95** |
| Streptopelia decaocto (Collared Dove) | 0.00 | 0.00 | 0.00 | 0.00 | 0.00 | 0.00 | **0.92** | **0.92** | **0.92** |
| Bombus pascuorum (Common Carder Bee) | 0.00 | 0.00 | 0.00 | **1.00** | 0.75 | 0.85 | **1.00** | **0.87** | **0.93** |
| Zootoca vivipara (Common Lizard) | 0.00 | 0.00 | 0.00 | 0.00 | 0.00 | 0.00 | **1.00** | **0.93** | **0.96** |
| Erithacus rubecula (Continental Robin) | **1.00** | 0.82 | 0.90 | **1.00** | 0.82 | 0.90 | **1.00** | **0.98** | **0.99** |
| Anthriscus sylvestris (Cow Parsley) | **1.00** | 0.27 | 0.43 | **1.00** | 0.27 | 0.43 | **1.00** | **0.28** | **0.44** |
| Prunella modularis (Dunnock) | 0.00 | 0.00 | 0.00 | 0.00 | 0.00 | 0.00 | **0.95** | **1.00** | **0.97** |
| Acer campestre (Field Maple) | 0.00 | 0.00 | 0.00 | 0.00 | 0.00 | 0.00 | 0.00 | 0.00 | 0.00 |
| Carduelis carduelis (Goldfinch) | **1.00** | 0.27 | 0.43 | **1.00** | 0.27 | 0.43 | 0.92 | 0.92 | 0.92 |
| Dendrocopos major (Great Spotted Woodpecker) | 0.00 | 0.00 | 0.00 | 0.00 | 0.00 | 0.00 | **1.00** | **0.97** | **0.99** |
| Parus major(Great Tit) | 0.00 | 0.00 | 0.00 | 0.00 | 0.00 | 0.00 | **0.97** | **0.97** | **0.97** |
| Chloris chloris (Greenfinch) | 0.00 | 0.00 | 0.00 | 0.00 | 0.00 | 0.00 | **0.95** | **1.00** | **0.97** |
| Passer domesticus (House Sparrow) | **1.00** | 0.35 | 0.52 | **1.00** | 0.43 | 0.60 | 0.97 | **0.85** | **0.90** |
| Corvus monedula (Jackdaw) | 0.00 | 0.00 | 0.00 | 0.00 | 0.00 | 0.00 | **1.00** | **0.94** | **0.97** |
| Pyrrhosoma nymphula (Large Red Damselfly) | 0.00 | 0.00 | 0.00 | 0.00 | 0.00 | 0.00 | **1.00** | **0.95** | **0.97** |
| Aegithalos caudatus (Long-Tailed Tit) | 0.00 | 0.00 | 0.00 | 0.00 | 0.00 | 0.00 | **1.00** | **0.95** | **0.97** |
| Pica pica (Magpie) | 1.00 | 0.44 | 0.61 | 1.00 | 0.44 | 0.61 | **0.88** | **0.85** | **0.87** |
| Phoxinus phoxinus (Minnow) | 0.00 | 0.00 | 0.00 | 0.00 | 0.00 | 0.00 | 0.00 | 0.00 | 0.00 |
| Lutra lutra (Otter) | 1.00 | 0.84 | 0.91 | 1.00 | 0.84 | 0.91 | **0.97** | **0.90** | **0.93** |
| Boloria euphrosyne (Pearl Bordered Fritillary) | 0.00 | 0.00 | 0.00 | 0.00 | 0.00 | 0.00 | **1.00** | **1.00** | **1.00** |
| Alca torda (Razorbill) | 0.00 | 0.00 | 0.00 | 0.00 | 0.00 | 0.00 | **0.93** | **0.76** | **0.84** |
| Riparia riparia (Sand Martin) | 0.00 | 0.00 | 0.00 | 0.00 | 0.00 | 0.00 | **0.88** | **0.69** | **0.78** |
| Argynnis paphia (Silver-Washed Fritillary) | 0.00 | 0.00 | 0.00 | 0.00 | 0.00 | 0.00 | **1.00** | **1.00** | **1.00** |
| Turdus philomelos (Song Thrush) | 0.00 | 0.00 | 0.00 | 1.00 | 0.02 | 0.05 | **0.94** | **0.94** | **0.94** |
| Sturnus vulgaris (Starling) | **1.00** | 0.08 | 0.15 | **1.00** | 0.08 | 0.15 | 0.97 | **0.86** | **0.91** |
| Passer montanus (Tree Sparrow) | 0.00 | 0.00 | 0.00 | 0.96 | 0.71 | 0.82 | **0.97** | **0.92** | **0.95** |
| Bombus lucorum (White-Tailed Bumble Bee) | 0.00 | 0.00 | 0.00 | **1.00** | 0.79 | 0.88 | **1.00** | **0.89** | **0.95** |
| Columba palumbus (Woodpigeon) | 0.00 | 0.00 | 0.00 | 0.00 | 0.00 | 0.00 | **0.90** | **0.91** | **0.94** |
| Troglodytes troglodytes (Wren) | **1.00** | 0.78 | 0.88 | **1.00** | 0.78 | 0.88 | **1.00** | **0.95** | **0.97** |
| Hedera helix (Ivy) | **1.00** | **0.18** | **0.31** | **1.00** | **0.18** | **0.31** | **1.00** | **0.18** | **0.31** |
| Sciurus carolinensis (Grey Squirrel) | **1.00** | 0.81 | 0.89 | **1.00** | 0.83 | 0.91 | **1.00** | **0.92** | **0.96** |

(*Continued*)

**Table 5.** (Continued)

| species | baseline | | | BOW (genus level) | | | BOW (class level) | | |
|---|---|---|---|---|---|---|---|---|---|
| | **p** | **r** | **F1** | **p** | **r** | **F1** | **p** | **r** | **F1** |
| Amanita rubescens (Blusher) | 0.00 | 0.00 | 0.00 | 0.00 | 0.00 | 0.00 | **1.00** | **0.94** | **0.97** |
| Cygnus atratus (Black Swan) | **1.00** | 0.77 | 0.87 | 0.96 | 0.84 | 0.89 | 0.90 | **0.90** | **0.90** |
| Morus bassanus (Gannet) | **1.00** | 0.69 | 0.82 | **1.00** | 0.69 | 0.82 | 0.97 | **0.94** | **0.95** |
| Branta leucopsis (Barnacle Goose) | 0.00 | 0.00 | 0.00 | 0.00 | 0.00 | 0.00 | **1.00** | **1.00** | **1.00** |
| AVERAGE | 0.29 | 0.16 | 0.20 | 0.39 | 0.23 | 0.27 | **0.86** | **0.76** | **0.79** |

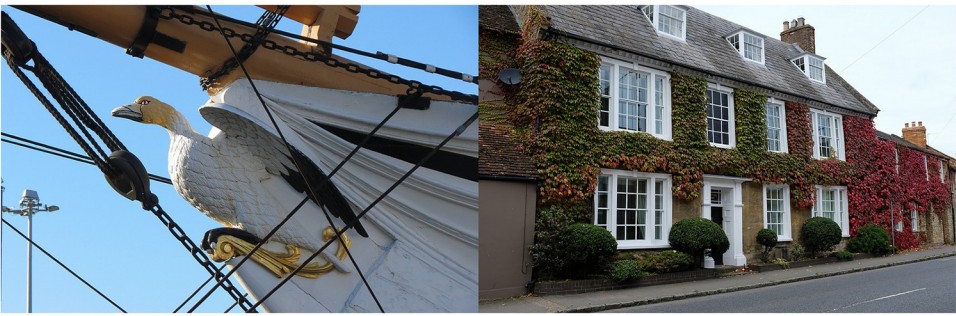

**Fig 4. Common cases of false positive and false negative: False positive for *Marus bassanus (Gannet)* (left) tags: Bird, goose, vehicle, tall ship and false negative for *Hedera helix (Ivy)* (right): Tags: Property, house, home, building, residential area, cottage, real estate, neighbourhood.**

Flickr records. Further, according to the conditions for three by three analysis comparisons, false negatives occur when a set of nine cells have at least one NBN record in the absence of any Flickr record for the given species. Therefore, for species where the number of NBN records is high and the number of Flickr records is low it is very likely that cells with no Flickr occurrences will be associated with cells containing NBN records (note however that for species that are well represented on Flickr this is less likely to be the case).

The highest precision and recall scores across both types of analysis are achieved for experiments performed with cell size 40km and no temporal constraints. The lowest results are achieved for experiments performed with a time constraint of twelve months which we attribute to lack of data on Flickr.

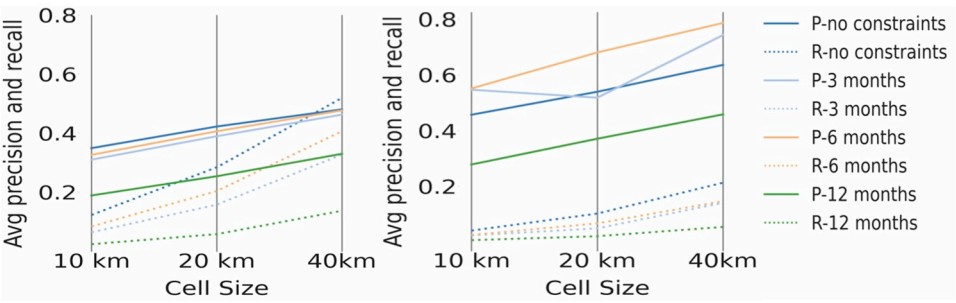

**Fig 5. Average precision and recall comparison per cell size and temporal restriction: One by one analysis (left), three by three analysis (right) where 'P' refers to precision and 'R' refer to recall.** In the figure 'no constraints' refers to the analyses performed with no temporal constraints.

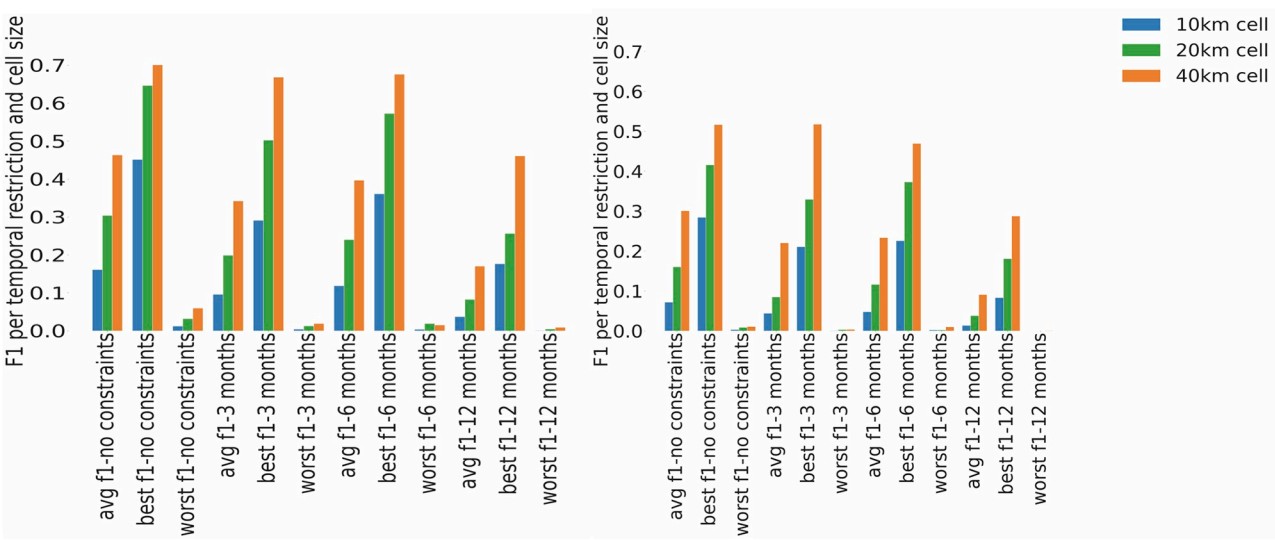

**Fig 6. Comparison of average, best, worst F1 measure values per temporal restriction and cell size: One by One analysis (left), three by three analysis(right).**

Precision tends to be higher than recall. This higher precision reflects the fact that most locations with Flickr occurrences also contain NBN occurrences. The low recall indicates that there are many locations with NBN observations but with no Flickr observations, leading as indicated above to false negatives. However, the recall value increases significantly as the cell size increases. The more relaxed spatial restrictions allow for what would otherwise be false negatives to become false positives as Flickr occurrences over a wider region are taken into account. Also, precision increases for bigger cell sizes for the converse reason of taking account of NBN occurrences over a wider region relative to a Flickr observation. This indicates that a grid split, consisting of 40km cells provides a better balance between precision and recall measures and thus can be regarded as more suitable for validating social network observations.

We calculated the best, worst and average F1 performance (see Fig 6), where best and worst were based on the average F1 scores for individual species for a particular cell size, while average F1 performance was across all species for the particular cell size. The average F1 measurement does not exceed 0.5. Best performing species have poorer F1 scores for cell sizes 10km and 20km and F1 score of 0.7 for the analysis performed on cell size 40km. F1-measure on average is higher when the analysis is performed with no temporal constraints. Further, the average F1 scores for the analysis conducted using a 12-month window are the lowest.

The results (see Figs 5 and 6) show that the Flickr dataset best reflects the NBN dataset on a purely spatial analysis with no time constraints. The comparison with a constraint that observations are within 12 months of each other gives the lowest results on all measures.

As indicated above, the overall comparison between the two datasets is notable for the highly unbalanced precision and recall scores. As these scores are averaged across all considered species, we investigated those species with precision and recall both being above 0.5, and we found 134 distinct such species. We found the average F1 score for the top 10 of these species with a 40km grid size to be 0.68 (see Table 6). As before the best results were obtained with no temporal constraints, though with a couple of exceptions for a 6 month temporal window. The best represented species on Flickr in comparison to NBN as represented in Table 6 are, with one exception, birds, most but not all of which are diurnal.

**Table 6. The top ten results with the highest f1-measure across all species.**

| Species name | Analysis type | Cell size | Precision | Recall | F1-measure |
|---|---|---|---|---|---|
| *Thymelicus sylvestris (Small Skipper)* | no constraints | 40 | 0.64 | 0.77 | 0.70 |
| *Strix aluco (Tawny Owl)* | no constraints | 40 | 0.65 | 0.76 | 0.70 |
| *Sitta europaea (Nuthatch)* | no constraints | 40 | 0.6 | 0.82 | 0.69 |
| *Primula veris (Cowslip)* | no constraints | 40 | 0.61 | 0.79 | 0.69 |
| *Aegithalos caudatus (Long-Tailed Tit)* | no constraints | 40 | 0.56 | 0.88 | 0.68 |
| *Botaurus stellaris (Bittern)* | no constraints | 40 | 0.61 | 0.76 | 0.68 |
| *Libellula depressa (Broad-Bodied Chaser)* | no constraints | 40 | 0.63 | 0.73 | 0.68 |
| *Sitta europaea (Nuthatch)* | 6 months | 40 | 0.62 | 0.74 | 0.68 |
| *Aegithalos caudatus (Long-Tailed Tit)* | 6 months | 40 | 0.59 | 0.78 | 0.67 |
| *Certhia familiaris (Treecreeper)* | no constraints | 40 | 0.56 | 0.83 | 0.67 |

**Invasive species for UK.** The average results for the invasive species demonstrate the same spatial and temporal patterns as the average results for the top 1500 species, presented in the previous section, i.e. best performance is for spatial analysis performed with 40 km grid cell size with no time constraints (Figs 7 and 8).

The average precision and recall across all species for each type of spatial and temporal constraint for one by one analysis is 0.4 (40%) for precision and 0.2 (20%) for recall (see Fig 7). The average precision and recall across all species for each type of spatial and temporal constraint for three by three analysis is 0.6 (60%) for precision and 0.1 (1%) for recall (see Fig 7).

The best represented invasive species on Flickr in comparison to NBN, with precision and recall both being above 0.5, are given in Table 7. Results are promising for these species as the F1-measure is on average 61.2%, specifically for representing spatial patterns on 40km cell size with no time constraints (see Table 7). There are six distinct species with the best performance among the invasive species (note that in Table 7 some species have multiple rows with different conditions of analysis). The species—*Branta canadensis (Canada Goose)* and *Sciurus carolinensis (Grey squirrel)* appear across the multiple categories of no temporal constraints, three months constraints and six months constraints. They are the best performing species in terms of having both precision and recall above 0.5 for multiple spatial and temporal restrictions and are the only species which have both precision and recall above 0.5 for cell size 20km. The best performance in terms of highest precision and highest F1 measure has been achieved for *Bubo bubo (Eurasian Eagle Owl)* with F1 = 0.71 and precision = 0.64 (with recall 0.79) based on a 40km cell size with no temporal constraints.

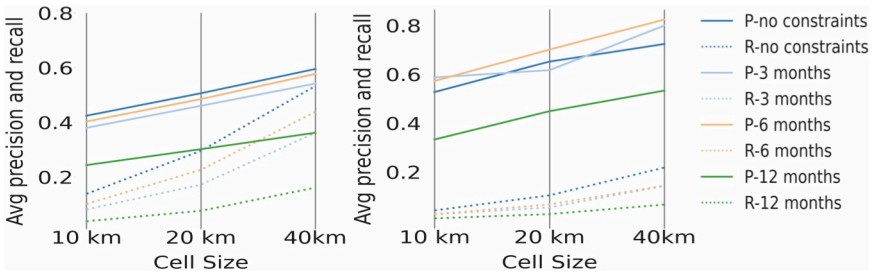

**Fig 7. Average precision and recall comparison per cell size and temporal restriction for invasive species: One by one analysis on the left, three by three analysis on the right, where 'P' refers to precision and 'R' refers to recall.** 'no constraints' refers to analyses performed with no temporal constraints.

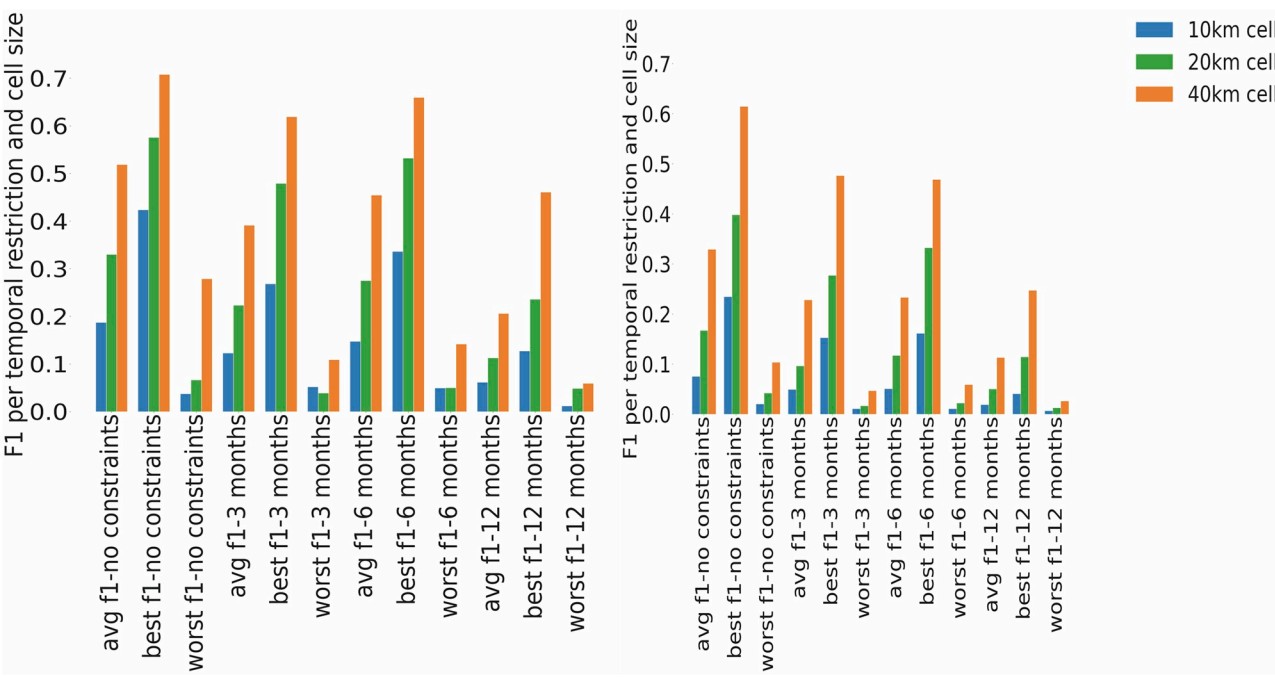

**Fig 8. Comparison of average, best, worst F1 measure values per temporal restriction and cell size for invasive species: The one by one analysis is on the left, the three by three analysis on the right.**

These results indicate that Flickr is a suitable platform for monitoring some types of invasive species for UK. Previous research on citizen science-based platforms for detecting invasive species has focused on invasive species mainly for US or Europe, rather than specifically the UK. Research by [28] showed that iNaturalist and eBird can be suitable for detecting early occurrences of invasive species. However, automatic extraction and analysis of this data as well as the need for organising volunteers are still remaining problems associated with the use of

**Table 7. Results for the invasive species where precision and recall are both above 0.5.**

| Species name | Analysis type | Cell size | Precision | Recall | F1-measure |
|---|---|---|---|---|---|
| *Branta canadensis (Canada Goose)* | no constraints | 40km | 0.55 | 0.80 | 0.65 |
| *Branta canadensis (Canada Goose)* | no constraints | 20km | 0.55 | 0.53 | 0.54 |
| *Cygnus atratus (Black Swan)* | no constraints | 40km | 0.59 | 0.79 | 0.68 |
| *Sciurus carolinensis (Grey squirrel)* | no constraints | 20km | 0.57 | 0.58 | 0.58 |
| *Sciurus carolinensis (Grey squirrel)* | no constraints | 40km | 0.59 | 0.80 | 0.68 |
| *Buddleja davidii (Buddleia)* | no constraints | 40km | 0.51 | 0.51 | 0.51 |
| *Bubo bubo (Eurasian Eagle Owl)* | no constraints | 40km | 0.64 | 0.79 | 0.71 |
| *Aix galericulata (Mandarin Duck)* | no constraints | 40km | 0.51 | 0.60 | 0.55 |
| *Cygnus atratus (Black Swan)* | 3 months | 40km | 0.55 | 0.54 | 0.55 |
| *Branta canadensis (Canada Goose)* | 3 months | 40km | 0.62 | 0.62 | 0.62 |
| *Sciurus carolinensis (Grey squirrel)* | 3 months | 40km | 0.59 | 0.63 | 0.62 |
| *Cygnus atratus (Black Swan)* | 6 months | 40km | 0.59 | 0.71 | 0.65 |
| *Branta canadensis (Canada Goose)* | 6 months | 40km | 0.59 | 0.69 | 0.64 |
| *Sciurus carolinensis (Grey squirrel)* | 6 months | 40km | 0.60 | 0.73 | 0.66 |
| *Bubo bubo (Eurasian Eagle Owl)* | 6 months | 40km | 0.55 | 0.75 | 0.64 |
| *Aix galericulata (Mandarin Duck)* | 6 months | 40km | 0.54 | 0.50 | 0.52 |

active citizen science approaches for detecting invasive species. Further, research on performing large-scale experiments on automatic approaches for identifying species from imagery data collected from citizen science portals revealed that automated species classification performs similarly and sometimes even better than manual annotations performed by citizen scientists [24, 25, 29]. In particular the authors of [24] presented a study encompassing 10,000 plant species while the authors of [25] performed a study for more than 5,000 categories of plants, animals, and fungi. These automated methods might in future facilitate better species identification on Flickr as well as other image resources.

Of the top 1500 most numerous species on NBN 90% were also found on Flickr, confirming that social media data can represent a wide range of species. A comparison between the two data collections on the diversity of species shows that NBN and Flickr datasets are similar on the class of species they represent. The best represented classes in both collections are the same with the top three being Insecta (Insects), Magnoliopsida (Plant class) and Aves (birds). Flickr has a good representation of flowering plants and garden and seabirds. Many Flickr uploads represent species that look attractive on photos and are easier to capture (i.e. they are diurnal, and/or are sessile) as well as being relatively common species.

Our image verification approach proved to work well on a large collection of species. The approach by [16] of exact match between the Google tags and species names may work for a small collection of well-known species for which the Google species labels tend to be more reliable, but not for a more extensive collection, including less well-known species, for which the Google label is liable to be more generic (i.e. providing the class or genus rather than the actual species name). In our approach, we use the taxonomy structure of the species to select relevant tags. Thus we verify images as genuine wildlife by matching the provided Flickr species name against the Google-provided class or the genus of the image content and all the tags lower down the classification hierarchy.

The spatial and temporal analyses for both case studies show that the Flickr dataset reflects the NBN dataset patterns best for experiments performed with cell size 40 km with no temporal constraints. The poorer results from the analysis performed with temporal constraints suggest that the Flickr dataset does not represent the temporal patterns for the species on NBN well. This is especially true for the yearly comparison between the two datasets (i.e. 12 month window).

The results of the precision calculations showed that there are 93 species for which precision is higher than 60%, for cell sizes 20km and 40km. This observation suggests that Flickr posts do present a potentially useful source of wildlife observations. However, the low recall value indicates that the Flickr data collection is less able to represent the full range of wildlife species in comparison to NBN. This is emphasised in the three by three analysis that gives the highest precision values, but provides the poorest recall. It should be remarked here that our scores for precision depend upon the quality of the NBN ground-truth data, which contains datasets collected through citizen science campaigns by non-professionals (as discussed in the Introduction). Therefore, it is quite possible that some of the Flickr observations classed here as false positive could actually be correct due to the absence of existing citizen science observations at the respective location. This suggests that social media data could be used to expand the range of species and locations observed where citizen science-related campaigns lack observations. Therefore, further studies on the 'false positive' results from Flickr in our analysis that evaluates against the NBN data, can be quite beneficial for revealing further whether social media data can be used to extend the range of species observed and facilitate more accurate analyses, such as into their migration patterns.

This problem leads to our next step, which is to conduct a similar larger scale study of the potential (beyond the relatively limited studies conducted to date) of other social networks

such as Twitter to determine whether they can also supplement traditional biodiversity data sources. Collecting social network data on a larger scale is a challenging task because most of the networks have restrictions on data access with thresholds on the amount of data that can be downloaded. A solution to this might be to look at how data from multiple social network sources can be combined for extracting wildlife data. It is also a strong motivation to apply and if possible improve upon methods for geocoding the many accessible social media posts that do not have GPS coordinates [30].

There is also scope to improve our image-verification method by looking for example at using a combination of inclusive and exclusive tags (i.e. tags used to consider a photo irrelevant) and through the development of more sophisticated computer vision methods for automated identification of individual species. We will also investigate methods of automatic verification that a social media posting is a genuine wildlife observation.

## Conclusions

This paper presented a large scale study exploring the potential of social media data to supplement citizen science datasets. In particular, we evaluated species distributions on Flickr relative to those submitted to the largest citizen science portal for the UK, the National Biodiversity Network (NBN) Atlas. Our study included the 1500 best represented species on NBN, and common invasive species within UK. We performed three types of analysis comparing the statistical, spatial, and temporal distribution of species on Flickr compared to NBN. Additionally, we presented a fully automated image verification method for identifying genuine species observations on Flickr, suitable for verifying large and diverse collections of species. The approach is based on the Google Cloud Vision API in combination with species taxonomic data to determine the likelihood that a mention of a species on Flickr represents a given species. This research showed that social network related data could offer a rich source of observation data for certain taxonomic groups, and/or as a repository for dedicated projects. In particular, spatial and temporal analysis suggest that the Flickr dataset best reflects the NBN dataset when considering a purely spatial distribution with no time constraints. The best represented species on Flickr in comparison to NBN are diurnal garden birds, as around 70% of the Flickr posts for them are valid observations relative to the NBN. In future, we plan to expand the study involving more social network sites, such as Twitter, and other citizen science portals. Additionally, we will consider improving the image verification approach by taking account of tags that might characterise individual species and hence indicate if photos are relevant to those species.

## Author Contributions

**Conceptualization:** Thomas Edwards, Christopher B. Jones, Sarah E. Perkins, Padraig Corcoran.

**Data curation:** Thomas Edwards.

**Formal analysis:** Thomas Edwards.

**Investigation:** Thomas Edwards.

**Methodology:** Thomas Edwards, Christopher B. Jones, Sarah E. Perkins, Padraig Corcoran.

**Project administration:** Christopher B. Jones.

**Software:** Thomas Edwards.

**Supervision:** Christopher B. Jones, Sarah E. Perkins, Padraig Corcoran.

**Validation:** Thomas Edwards, Sarah E. Perkins.

**Writing – original draft:** Thomas Edwards, Christopher B. Jones, Sarah E. Perkins, Padraig Corcoran.

**Writing – review & editing:** Thomas Edwards, Christopher B. Jones, Sarah E. Perkins, Padraig Corcoran.

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
