## [Decision Letter · Decision Letter 0]

7 Apr 2021

PONE-D-21-05380

Passive citizen science: the role of social media in wildlife observations

PLOS ONE

Dear Dr. Edwards,

Thank you for submitting your manuscript to PLOS ONE. After careful consideration, we feel that it has merit but does not fully meet PLOS ONE’s publication criteria as it currently stands. Therefore, we invite you to submit a revised version of the manuscript that addresses the points raised during the review process.

We look forward to receiving your revised manuscript.

Kind regards,

Daniel de Paiva Silva, Ph.D.

Academic Editor

PLOS ONE

Journal Requirements:

2. In your Methods, please provide a statement indicating that the use of this dataset was done in compliance to the National Biodiversity Network and Flickr Terms and Conditions and our requirements for this type of study (https://journals.plos.org/plosone/s/submission-guidelines#loc-personal-data-from-third-party-sources).

Additional Editor Comments (if provided):

Dear Edwards et al.,

Greetings! After the evaluation by two independent reviewers, both of them found that your manuscript has great publication chances. Still, improvements are always required. Please take a closer look on what both reviewers indicated about your manuscript, and arrange the necessary improvements they asked. By the time you are required to resubmit a new version of your text, please do not forget to prepare a rebuttal letter indicating all the changes you did and justifying all of those you did not.

Considering the pandemic scenario, a three-months period (resubmission due on July 3rd 2021)may be enough for you to prepare all the changes. If not, please let me know how could we arrange a better schedule for you. In case you are able to resubmit earlier then the provided due, do not hesitate to do it.

Sincerely,

Daniel Silva

Reviewers' comments:

Reviewer's Responses to Questions

**Comments to the Author**

1. Is the manuscript technically sound, and do the data support the conclusions?

Reviewer #1: Partly

Reviewer #2: Yes

2. Has the statistical analysis been performed appropriately and rigorously? 

Reviewer #1: Yes

Reviewer #2: Yes

3. Have the authors made all data underlying the findings in their manuscript fully available?

Reviewer #1: Yes

Reviewer #2: Yes

4. Is the manuscript presented in an intelligible fashion and written in standard English?

Reviewer #1: Yes

Reviewer #2: Yes

5. Review Comments to the Author

Reviewer #1: In the manuscript “Passive citizen science: the role of social media in wildlife observations”, the authors assess the usefulness of social media data (images on Flickr) for providing species occurrence data. To do this, they extract geolocation and temporal information from Flickr photos and, treating this as a species occurrence record, compares this to records on the National Biodiversity Network (NBN) Atlas. They use Google Cloud Vision API and species’ taxonomic classifications to verify the species tagged in the Flickr images. After evaluating 1500 UK species, as well as numerous invasive species, they found that the Flickr data best represents the NBN data when considered over large scales (40km grids) without temporal constraints. The authors conclude that Flickr can potentially provide useful data on species’ distributions, especially when combined with existing biodiversity data.

Overall, the manuscript is succinct and well written, and I thoroughly enjoyed reading it. The topic of the study is interesting and timely. Traditional citizen science has gained traction in recent years as a reliable and efficient means of gathering wildlife data. Therefore, it seems only fitting to explore this further by evaluating the potential contributions of ‘passive citizen science’ and specifically social media.

The authors provide a good overview of the current state of the literature and identifies a clear research gap. In general, I found the methods to be appropriate and relatively clear, but some sections would benefit from further clarification (please see specific comments below). Likewise, I found the discussion to be sound based on the presented results (which were generally clear but inconsistent in some areas; see comments below). However, most, if not all, citizen science datasets are subject to questions surrounding data quality and the Flickr data is no exception. In this case, I think that there are some caveats to the analyses that need to be addressed.

INTRODUCTION:

Lines 78-83: The authors refer to a paper by ElQadi et al. (2017), describing the methods in this paper as “exact matching between species names … and the labels (tags) returned by Google’s Reserve Image Search”. They state that this can lead to “false negatives where the Google API provides only a more general label or another name for the species”. While I concur that an “exact matching” method would result in false negatives, I do not completely agree with the authors’ interpretation of ElQadi et al.’s methods.

ElQadi et al. searched for images on Flickr using species’ scientific and common names. These images were then given a text tag using Google Reverse Image Search. They used tag frequency to filter out irrelevant images, such that tags with low frequency (regardless of their relevance) were excluded. Consequently, images with relevant tags, including tags of the species’ scientific or common names, were sometimes nonetheless excluded if the tags were infrequent (see paragraph 2 of their discussion). Yes, ElQadi et al.’s method still results in false negatives, but it is not due to “exact matching” between species names and Google’s tags. “Exact matching” implies that an image was deemed relevant (and thus included after the filtering step) only if the Google tags exactly matched the species names, which was not the case. For example, in ElQadi et al.’s study, Flickr images of the Blue banded bee (Amegilla cingulata) that were tagged by Google as “Amegilla cingulata” and “blue banded bee” were nevertheless excluded after their filtering step because of the relative infrequency of these tags. In other words, false negatives occurred even when Google API tagged the exact species name. Furthermore, photos of blue banded bees that were given frequent general tags, such as “insect” and “bee”, were included after the filtering step. Thus, false negatives do not always occur when general labels are provided. The authors’ description should therefore be clarified. Further mentions of the “exact-match” method in relation to ElQadi et al.’s paper (Lines 278 and 393) should also be clarified.

METHODS:

Line 135: To conduct their temporal analyses, the authors downloaded date information (the date that the image was uploaded) from each Flickr image. However, I expect some images would have been uploaded some time (potentially a long time) after being taken. Is there any way of knowing when the photo was actually taken, as opposed to uploaded? This information might not be readily available, but if there are frequent discrepancies between when photos were taken and when they were uploaded, it is difficult to have confidence in the results of the temporal analyses (seasonal, half yearly, and yearly patterns). If this information is not available, the authors should at least address how this might have affected the results of the temporal analyses.

Lines 171-174: This is a fair statement, but I am wondering what happens if the Flickr image features another species of grass that has been incorrectly tagged on Flickr as Timothy grass? Would the coarse match still be successful in this case? If so, how often might this occur and how might this impact the results? While I think the authors’ image verification technique goes some way in improving previous methods, it appears that the reliability of Flickr-derived species occurrence data still depends significantly on the ability of the Flickr users (i.e., the ‘passive citizen scientists’) to correctly assign species name tags to Flickr images. The authors should discuss this if this is the case.

RESULTS AND DISCUSSION:

Lines 277-279: What were the 10 species and how were these chosen? I think it would be ideal to choose the species randomly, but either way, a justification should be provided. My concern is that the image verification approach may perform better for certain species or taxonomic groups than others. Given this, I think it would be best to evaluate the performance of the verification technique using the full range of species available in the dataset.

General comment: The authors did a good job addressing the key points in this section. I would have liked some more discussion around why the authors thought that constraining the analyses to a 12-month window led to poorer results than a 3 or a 6-month window. On Lines 323-325, the authors attribute the poor results in the 12-month analyses to a “lack of data on Flickr”. Intuitively, should this phenomenon (i.e., the lack of data) not be even more pronounced when constraining the analyses to an even shorter time frame (i.e., 3 months and 6 months)?

FIGURES:

Overall, the figures were relevant and suitable for the data. I particularly appreciate the inclusion of Figure 1. The flow chart and the matching sub-headings in the methods were helpful.

MINOR ISSUES:

Lines 33-34: The authors state that “geotags given to photos on social media sites … are assigned automatically by GPS location systems”. Whilst this may be true of photos taken on a location enabled device, there would be cases where automatic geotags are not available (e.g., on a device without a GPS system), or where the geotags are low in accuracy. Does Flickr allow users to manually add or edit location data? If so, I think this needs to be stated, either here or in the methods.

Line 126: What is meant by “exact search”? It would also be good to get an idea of how many records included irrelevant species in any given search.

Line 136: It would be helpful to add more detail about the Flickr tags. Are they manually entered by the user/uploader, or are they automatically assigned by the computer, or both? How many tags are allowed? The authors did an excellent job of explaining the Google Cloud Vision API tags (lines 151-157), so something like this, explaining the Flickr tags and clearly distinguishing them from the Google API tags, would be great. I initially had some difficulty following the authors’ methods in relation to which tag they were referring to (i.e., the Flickr tags or the Google API tags).

Line 169: The authors write, “Using the exact matching on the Flickr label of Adder would not have found any match”. This is unclear to me. Why would the Flickr label “Adder” not have found an exact match when “Adder” is indeed one of the species’ common names in the NBN taxonomic classification (Figure 2)? Is it because Google API does not label the image of the adder as “Adder”? Please clarify.

Lines 177-178: “… for a 100 randomly selected instances”. Either a word is missing here, or the “a” should be removed.

Lines 229-230: “… and a 100% of species”. Either a word is missing here, or the “a” should be removed.

Lines 263-266: This sentence would be clearer if these species were listed in order based on their Flickr species count, starting with the best represented species.

Lines 278-279: “… for a 1000 randomly selected instances”. Either a word is missing here, or the “a” should be removed.

Line 307: Rather than writing “correct”, I suggest simply stating that 38% of the Flickr data were also reflected in the NBN data. These mean slightly different things as an absence of NBN data from a location (in the presence of Flickr data) does not necessarily mean that the Flickr identifications were incorrect. Rather, the species may have just never been recorded there previously (which the authors note on Lines 415-416). In fact, these “false positives” could actually be interesting datapoints as potential range expansions, especially given invasive species are a substantial focus of this manuscript.

Line 331: Did the authors mean to write “true positives” rather than “false positives”? False positives were not used to calculate recall.

Line 372: The authors mentioned seven distinct species but presented only six in the table. Please clarify.

Line 407: It would be good to provide a number or percentage here alongside the claim of “a large number”.

Table 1: I think there is a missing row here. The table legend (and Line 234) refers to the “top 10” most frequently recorded species on Flickr, but only 9 species are shown in the table.

Tables 1 & 2: Please check the Flickr count number for Erithacus rubecula (Continental Robin). The count is given as 19,248 in Table 1 but 2,786 in Table 2. Meanwhile, the NBN count is the same in both tables. If this is not an error, please provide an explanation as to why these differ.

Table 3: The table legend states, “Species occurrences … with number of species above 100”. Did the authors mean to write, “with number of occurrences above 100”?

Figures 4-7: I suggest making the left and right panels/graphs the same width, especially in Figures 4 and 6, so that the gradients of the lines can be compared.

Reviewer #2: This paper has the aim to put into value the big amount of biodiversity data hidden in social media. Specifically, they put to the test the photo base on Flickr. Flickr data were evaluated relative to the National Biodiversity Network (NBN) Atlas, the largest collection of species distribution data in the UK. They used an innovative image verification technique that uses the Google Cloud Vision API in combination with species taxonomic data to determine the likelihood that a mention of a species on Flickr represents a given species. Authors found that data are useful for representing some common and diurnal species on a space scale but not on a temporal scale. This work is an interesting and useful contribution to better understanding the value of biodiversity data from social media to contribute to science. However, there are some recommendations to get this article improved for publication.

Introduction

First, I recommend better explain the relation of this manuscript in the context of citizen science on introduction. The definition of citizen science still an open discussion today and this concept of “passive citizen science” could be controversial. I propose to better definition of passive citizen science immerses in a framework of citizen science definition.

Material and Methods

I consider the material and methods are ok. However, I recommend better explain how the NBN database is performed. Where data came from?

Results and discussion

Results are generally well explained and represented. But I consider some parts of the results are weakly discussed and more scientific references would be welcome. You could discuss also the potential of this data, for example comparing it with other more active citizen science methods based on submitted photographs (e.a, iNaturalist).

More specific comments:

Line 3: Please check the references on the whole text. Is this the correct format for the journal?

Lines 54-56: Please, better explain how the NBN database was made.

Line 62: Please, abbreviate “National Biodiversity Network Atlas”

Lines 371-376: Discuss these results. The occurrence of common species in citizen science data is registered in some articles you could refer to.

Lines 413-416: I do not understand this sentence. NBN database is a citizen science database?

Line 416: there is a parenthesis

6. PLOS authors have the option to publish the peer review history of their article (what does this mean?). If published, this will include your full peer review and any attached files.

Reviewer #1: No

Reviewer #2: **Yes: **Maria Isabel Hermoso Beltrán

---

## [Author Response · Author response to Decision Letter 0]

2 Jul 2021

We would like to thank the editor and the reviewers for their insightful comments. We have addressed all suggestions in the revised manuscript. Additionally, we have responded to the reviewer's comments below. 

 Reviewer # 1

1) Lines 78-83: The authors refer to a paper by ElQadi et al. (2017), describing the methods in this paper as “exact matching between species names ... and the labels (tags) returned by Google’s Reserve Image Search”. They state that this can lead to “false negatives where the Google API provides only a more general label or another name for the species”. While I concur that an “exact matching” method would result in false negatives, I do not completely agree with the authors’ interpretation of ElQadi et al.’s methods. ElQadi et al. searched for images on Flickr using species’ scientific and common names. These images were then given a text tag using Google Reverse Image Search. They used tag frequency to filter out irrelevant images, such that tags with low frequency (regardless of their relevance) were excluded. Consequently, images with relevant tags, including tags of the species’ scientific or common names, were sometimes nonetheless excluded if the tags were infrequent (see paragraph 2 of their discussion). Yes, ElQadi et al.’s method still results in false negatives, but it is not due to “exact matching” between species names and Google’s tags. “Exact matching” implies that an image was deemed relevant (and thus included after the filtering step) only if the Google tags exactly matched the species names, which was not the case. For example, in ElQadi et al.’s study, Flickr images of the Blue banded bee (Amegilla cingulata) that were tagged by Google as “Amegilla cingulata” and “blue banded bee” were nevertheless excluded after their filtering step because of the relative infrequency of these tags. In other words, false negatives occurred even when Google API tagged the exact species name. Furthermore, photos of blue banded bees that were given frequent general tags, such as “insect” and “bee”, were included after the filtering step. Thus, false negatives do not always occur when general labels are provided. The authors’ description should therefore be clarified. Further mentions of the “exact-match” method in relation to ElQadi et al.’s paper (Lines 278 and 393) should also be clarified.

---- Response: Thank you for identifying the problem. We agree with your interpretation of the approach and have revised the relevant text. The way we interpret it and why we refer to it as an exact match is because of the following: 

The authors (ElQadi et al) use the Google Reverse Image Search, which can return multiple tags per photo. The authors order all tags per species in descending order of frequency. Then, they manually identify which are the species-relevant and irrelevant tags, among the most frequent ones, that can help indicate which photos are true representations of the given species. This manual selection of relevant tags per species we consider an ‘exact match’ between the specific species and what would be the most relevant tags for this specific species. Instead, we use the NBN species classification framework to identify species-relevant tags. Despite the benefits of such an approach especially when photos need to be evaluated only for a couple of species, we consider it unsuitable for larger collections where the manual selection of relevant tags per species will be a time-consuming process. We agree that after the many iterations of the paper, we have excluded this information. Further, we agree that the baseline approach based on exact matching between species names needs further clarifications. The exact match approach is inspired by ElQadi's work as verification is performed on species-level. However, we wanted to emphasize on the benefits of BoW approach over fully automatic exact match approach which potentially might give higher precision but lower recall. In this way, we compare two fully automated approaches, the one having the advantage of potentially giving higher recall and the other higher precision. As results showed, however, the BoW approach led to higher recall value without affecting the precision measure. We have included these clarifications in the paper (lines 125 - 134). 

2) Line 135: To conduct their temporal analyses, the authors downloaded date information (the date that the image was uploaded) from each Flickr image. However, I expect some images would have been uploaded some time (potentially a long time) after being taken. Is there any way of knowing when the photo was actually taken, as opposed to uploaded? This information might not be readily available, but if there are frequent discrepancies between when photos were taken and when they were uploaded, it is difficult to have confidence in the results of the temporal analyses (seasonal, half yearly, and yearly patterns). If this information is not available, the authors should at least address how this might have affected the results of the temporal analyses.

---- Response: Flickr API allows downloading two dates associated with a photo, 'taken date' and 'posted date'. 'Taken date' represents the time at which the photo was taken. This information is extracted from the EXIF date if available, otherwise the 'taken' date is set to the 'posted date'. 'Posted date' represents the time at which the photo was uploaded to Flickr. For performing the temporal analysis, we use the 'taken date'. However, early observations showed that 'posted date' and 'taken date' do not differ more than 3 months for the majority of photos. Further, photos are often uploaded on Flickr directly after they have been taken. We have clarified this in the paper and made a clear distinction between the two types of dates and which date type we used for the temporal analysis (lines 190 - 199).

3) Lines 171-174: This is a fair statement, but I am wondering what happens if the Flickr image features another species of grass that has been incorrectly tagged on Flickr as Timothy grass? Would the coarse match still be successful in this case? If so, how often might this occur and how might this impact the results? While I think the authors’ image verification technique goes some way in improving previous methods, it appears that the reliability of Flickr-derived species occurrence data still depends significantly on the ability of the Flickr users (i.e., the ‘passive citizen scientists’) to correctly assign species name tags to Flickr images. The authors should discuss this if this is the case.

---- Response: Thank you for this question. The coarse match might not be successful in the scenario you described. However, a main drawback of previous image verification approaches which we aim to address, is the inability of these methods to scale to large collections of species and the need for manual or semi-automatic verification. A main reason for this is that some species have very similar characteristics and cannot be accurately distinguished by image recognition models. Our image verification approach is the first attempt, to the best of our knowledge, that addresses this issue and provides a way for verifying large and diverse image-related species data fully automatically. The motivation of using a fully automated approach is the need for fast and less resource consuming methods to efficiently verify large collections of species-related images. We hypothesize that a coarse match will help improve the coverage of automated methods (recall measure) without affecting their precision. We include a more detailed description of the image validation approach and discussion in the paper (lines 201 - 235, lines 390 - 417). Additionally, we addressed the concern that class-level BoW approach might be too coarse and lead to lower precision by also performing evaluation of BoW approach on genus level. In this way, we compare three evaluation approaches -- exact match (i.e., perform exact match between species names and species name given per Flickr photo), class level BOW approach (i.e., consider names following down from the class of the species), and genus level BOW approach (i.e., consider names following down from the genus of the species). This allows us to judge which approach is more suitable for evaluating diverse collections of species without affecting the precision of approaches. Results from this comparison are presented in Table 5. Finally, we performed manual analysis on the false negative and false positive cases returned by BoW approach, and we found that the most common cases of false positives are photos that include an artificial representation of a species and the most common cases of false negatives are photos which include the species, but the focus of display is another object. We did not encounter the problems you mentioned during our evaluation, but we do agree these cases might exist. Our main aim was to provide a fully automated approach for image verification that scales well for large collections of species. We plan in future to improve our method by training the image verification models on wildlife related data and use a matching algorithm that incorporates a combination of multiple labels, representing more species characteristics such as colour patterns and size. We believe that this will help provide a more fine-grain distinguishment between the different species. 

4) Lines 277-279: What were the 10 species and how were these chosen? I think it would be ideal to choose the species randomly, but either way, a justification should be provided. My concern is that the image verification approach may perform better for certain species or taxonomic groups than others. Given this, I think it would be best to evaluate the performance of the verification technique using the full range of species available in the dataset.

---- Response: Thank you for the suggestion. We have extended the evaluation to 50 randomly chosen species, each associated with 40 images which gives us in total 2000 images (lines 272 - 274). Additionally, we compared the class-level BOW approach to exact match and genus-level BOW approach (see response to question 3)). Results and discussion of the new results are given in Section ‘Flickr data verification’. We could not perform evaluation of the entire image collection as it is a very time-consuming process which would involve the careful observation of more than 1,000,000 images. 

5) I would have liked some more discussion around why the authors thought that constraining the analyses to a 12-month window led to poorer results than a 3 or a 6-month window. On Lines 323-325, the authors attribute the poor results in the 12-month analyses to a “lack of data on Flickr”. Intuitively, should this phenomenon (i.e., the lack of data) not be even more pronounced when constraining the analyses to an even shorter time frame (i.e., 3 months and 6 months)?

---- Response: Thank you for the question. In our experiments we identify seasonal (3 month window) and half yearly patterns (6 month window) such as seasonal migrations which might be affected mainly by weather changes rather than the year they appear in. Therefore, for these time shorter window experiments, we ignore the year and perform analysis across all years instead. For instance, for the 6 month window, we consider all observations recorded between January - June across the years 2006 - 2017 (the years for which we collected Flickr data). Therefore, observing seasonal patterns across all years will result in more data than simply observing yearly patterns. We clarify this in the paper (lines 321 - 334). 

6) Lines 33-34: The authors state that “geotags given to photos on social media sites ... are assigned automatically by GPS location systems”. Whilst this may be true of photos taken on a location enabled device, there would be cases where automatic geotags are not available (e.g., on a device without a GPS system), or where the geotags are low in accuracy. Does Flickr allow users to manually add or edit location data? If so, I think this needs to be stated, either here or in the methods.

---- Response: Thank you for the suggestion. We consider only posts where coordinates are extracted from a GPS-enabled device which is either the device used to upload the photo, or the device used to take the photo. We ignore posts associated only with user-provided locations. We included this in the paper (lines 197 - 199).

7) Line 126: What is meant by “exact search”? It would also be good to get an idea of how many records included irrelevant species in any given search.

 ---- Response: Exact match refers to a search term match where the search term fully matches the species name associated with the record. For instance, there is an exact match between the search term ‘Canada Goose’ and the species NBN record name ‘Canada Goose’. Therefore, this exact match will return results relevant only to Canada Goose species. However, NBN Atlas supports partial match where using a search term such as ‘Grey Squirrel’ will also return records for ‘Red Squirrels’ as the two species names partially match (squirrel). In order to avoid this, once we have performed the search, we remove the species records irrelevant to the search term. We did not keep records of these species. We clarify this in the paper (lines 175 - 182).

8) Line 136: It would be helpful to add more detail about the Flickr tags. Are they manually entered by the user/uploader, or are they automatically assigned by the computer, or both? How many tags are allowed? The authors did an excellent job of explaining the Google Cloud Vision API tags (lines 151-157), so something like this, explaining the Flickr tags and clearly distinguishing them from the Google API tags, would be great. I initially had some difficulty following the authors’ methods in relation to which tag they were referring to (i.e., the Flickr tags or the Google API tags).

 ---- Response: Thank you for the suggestion. Flickr tags were manually entered by the user where a user is allowed to write 75 tags maximum per photo. We acknowledge that the terminology we used is confusing and that the Flickr validation approach needed further explanations. Therefore, we included Table 1 describing main concepts used in the Flickr verification method and Fig 2 providing an overview of the image validation approach. A more detailed description of the methodology is given in lines 201 - 237.

9) Line 169: The authors write, “Using the exact matching on the Flickr label of Adder would not have found any match”. This is unclear to me. Why would the Flickr label “Adder” not have found an exact match when “Adder” is indeed one of the species’ common names in the NBN taxonomic classification (Figure 2)? Is it because Google API does not label the image of the adder as “Adder”? Please clarify.

 ---- Response: Yes, it is as the reviewer suspects. We perform a match between the NBN names associated with each species represented in the Flickr photos and Google Cloud Vision API labels returned per photo in order to identify whether the species names given by the Flickr users are correctly representing the species displayed on the photos. Further, very often Flickr photos of the species adder are associated with the tag “Adder” while Google Cloud Vision API returns the label “Viper” for the same photo. Thus, if we simply perform an automatic match between the Google Cloud Vision API label “Viper” and the Flickr tag “Adder” there will be no match and the photo will be counted mistakenly as a false representation of Adder. We overcome this problem by using the NBN classification names for the species Adder. This helps expand the name list of the given species with other names used for adder such as “viper” which is the label used by Google Cloud Vision API. We further clarify the process of image verification in the paper (lines 201 - 237). 

10) Lines 177-178: “... for a 100 randomly selected instances”. Either a word is missing here, or the “a” should be removed.

 ---- Response: Thank you for pointing out the mistake. We removed “a”.

11) Lines 229-230: “... and a 100% of species”. Either a word is missing here, or the “a” should be removed.

 ---- Response: Thank you for pointing out the mistake. We removed “a”.

12) Line 307: Rather than writing “correct”, I suggest simply stating that 38% of the Flickr data were also reflected in the NBN data. These mean slightly different things as an absence of NBN data from a location (in the presence of Flickr data) does not necessarily mean that the Flickr identifications were incorrect. Rather, the species may have just never been recorded there previously (which the authors note on Lines 415-416). In fact, these “false positives” could actually be interesting data points as potential range expansions, especially given invasive species are a substantial focus of this manuscript.

 ---- Response: Thank you for the observation. We updated line 307 and now it only states 38%. Additionally, we included more discussion about the false positive observations on lines 557 - 562. 

13) Line 331: Did the authors mean to write “true positives” rather than “false positives”? False positives were not used to calculate recall.

 ---- Response: Yes, false positives are not used to calculate recall. Instead, recall is calculated using the number of false negatives while precision is calculated using the number of false positives. When we have a higher number of false negatives the recall value decreases while when we have a higher number of false positives the precision value decreases. Precision is used to show how many selected instances are relevant while recall is used to show how many of the total number of relevant instances are selected. The statement ‘The more relaxed spatial restrictions allow for what would otherwise be false negatives to become false positives as Flickr occurrences over a wider region are taken into account’ mean that the number of false negatives will decrease and consequently the recall value will increase. However, this will lead to an increase of the number of false positives which will lead to lower precision value. 

14) Line 372: The authors mentioned seven distinct species but presented only six in the table. Please clarify.

 ---- Response: Thank you for pointing out the mistake. We corrected it in the paper. There are six such species.

15) Line 407: It would be good to provide a number or percentage here alongside the claim of “a large number”.

 ---- Response: Thank you for the suggestion. We included the exact number in the paper which is 93 species (line 546).

16) Table 1: I think there is a missing row here. The table legend (and Line 234) refers to the “top 10” most frequently recorded species on Flickr, but only 9 species are shown in the table.

 ---- Response: Thank you for pointing out the mistake. We fixed it and included the 10th species in the table.

17) Tables 1 & 2: Please check the Flickr count number for Erithacus rubecula (Continental Robin). The count is given as 19,248 in Table 1 but 2,786 in Table 2. Meanwhile, the NBN count is the same in both tables. If this is not an error, please provide an explanation as to why these differ.

 ---- Response: Thank you for pointing out the mistake. We fixed it in the paper. The number of occurrences on Flickr for Erithacus rubecula (Continental Robin) is 19,248.

18) Table 3: The table legend states, “Species occurrences ... with number of species above 100”. Did the authors mean to write, “with number of occurrences above 100”?

 ---- Response: Thank you for the correction. Yes, we meant “with number of occurrences above 100”. We fixed these issues.

19) Figures 4-7: I suggest making the left and right panels/graphs the same width, especially in Figures 4 and 6, so that the gradients of the lines can be compared.

 ---- Response: Thank you for pointing out this inconsistency. We addressed it. 

Reviewer # 2

1) I recommend better explain the relation of this manuscript in the context of citizen science on introduction. The definition of citizen science still an open discussion today and this concept of “passive citizen science” could be controversial. I propose to better definition of passive citizen science immerses in a framework of citizen science definition.

 ---- Response: Thank you for your suggestion. We addressed this suggestion in the paper (lines 12 -15, lines 32 - 41)

2) I recommend better explain how the NBN database is performed. Where data came from?

 ---- Response: Thank you for your suggestion. We addressed this suggestion in the paper (lines 92 - 94; lines 97 - 104).

3) But I consider some parts of the results are weakly discussed and more scientific references would be welcome. You could discuss also the potential of this data, for example comparing it with other more active citizen science methods based on submitted photographs (e.a, iNaturalist).

 ---- Response: Thank you for the suggestion. We have included a discussion of iNaturalist platform and other approaches, compared to social network sources in the Introduction (lines 18 - 25 and 67-87) and extended the discussion of our results to include citation of other papers (lines 507 - 521). In future, we will further extend comparison between passive citizen science and active citizen science methods by also including data from iNaturalist. The reason for using NBN for performing our analysis is the wide usage of the website for ecology research and the fact that it holds the largest collection of citizen science observations of species for the UK (we focus on UK-based analysis). Further, NBN provides programmable APIs which help fast and efficient data analysis to be performed on a larger scale.

4) Line 3: Please check the references on the whole text. Is this the correct format for the journal?

 ---- Response: We checked our style files and used the latex template for PLOS ONE journal available at:https://journals.plos.org/plosone/s/latex and the bibliography style provided as part of it. Further, we used guideline pages available at: https://journals.plos.org/plosone/s/file?id=wjVg/PLOSOne_formatting_sample_main_body.pdf and https://journals.plos.org/plosone/s/file?id=ba62/PLOSOne_formatting_sample_title_authors_affiliations.pdf. However, if we have missed some information and we still need to update the format of the manuscript we are happy to do that.

5) Lines 54-56: Please, better explain how the NBN database was made.

 ---- Response: We have included explanations in the paper (lines 75 - 81).

6) Line 62: Please, abbreviate “National Biodiversity Network Atlas”

 ---- Response: Thank you for pointing out the inconsistencies. We addressed this in the paper (lines 92 - 95).

7) Lines 371-376: Discuss these results. The occurrence of common species in citizen science data is registered in some articles you could refer to.

 ---- Response: We have extended discussion in the paper (lines 507 - 521). Additionally, in the `Introduction’ section, we extended the discussion of previous research on using social media as a source of wildlife data (lines 67 - 87). 

8) Lines 413-416: I do not understand this sentence. NBN database is a citizen science database?

 ---- Response: Yes, NBN holds wildlife observation data gathered by non-professionals but citizen scientists. Therefore, it is possible that the records uploaded on NBN can be enhanced by species locations that are correctly captured by Flickr users. We have clarified this further in the paper (lines 553 - 554)

9) Line 416: there is a parenthesis

 ---- Response: Thank you for the correction. We addressed in the paper.

Sincerely,

On behalf of all authors, 

Thomas Edwards

---

## [Decision Letter · Decision Letter 1]

16 Jul 2021

Passive citizen science: the role of social media in wildlife observations

PONE-D-21-05380R1

Dear Dr. Edwards,

We’re pleased to inform you that your manuscript has been judged scientifically suitable for publication and will be formally accepted for publication once it meets all outstanding technical requirements.

Kind regards,

Daniel de Paiva Silva, Ph.D.

Academic Editor

PLOS ONE

Additional Editor Comments (optional):

Dear Edwards et al.,

after a final review round by one of the reviewers who previously reviewed your original MS, I believe you text is now suitable (and formally accepted) for publication in PLoS One! Congratualations!

Sincerely,

Daniel Silva, Ph.D.

Reviewers' comments:

Reviewer's Responses to Questions

**Comments to the Author**

1. If the authors have adequately addressed your comments raised in a previous round of review and you feel that this manuscript is now acceptable for publication, you may indicate that here to bypass the “Comments to the Author” section, enter your conflict of interest statement in the “Confidential to Editor” section, and submit your "Accept" recommendation.

Reviewer #1: All comments have been addressed

2. Is the manuscript technically sound, and do the data support the conclusions?

Reviewer #1: Yes

3. Has the statistical analysis been performed appropriately and rigorously? 

Reviewer #1: Yes

4. Have the authors made all data underlying the findings in their manuscript fully available?

Reviewer #1: Yes

5. Is the manuscript presented in an intelligible fashion and written in standard English?

Reviewer #1: Yes

6. Review Comments to the Author

Reviewer #1: I thank the authors for their careful and considered responses to my concerns and queries. These have been satisfactorily addressed. The authors’ revisions have improved the clarity of the manuscript and I specifically appreciate their efforts to clarify their methods. The additional information should provide readers with sufficient information to make an informed judgement on the strengths and potential limitations of the current study.

Overall, this is a great manuscript that demonstrates the potential of social media to provide valuable biodiversity data.

I noticed just a few additional (very minor) issues which can be easily addressed:

Line 324: Are you missing a word in “This allows to identify…”?

Line 345: Please check the date range (2006-2018). Is this date range meant to be different to the temporal analysis date range (2004-2017; Line 327)?

Line 397: Are you missing a “with” in “associated a given species”?

Line 513: Please remove the duplicate “that”.

Line 589: Should “observation” be “observations”?

Figure 2: “Coarse match b-n NBN Google labels” – what is “b-n”? Between?

7. PLOS authors have the option to publish the peer review history of their article (what does this mean?). If published, this will include your full peer review and any attached files.

Reviewer #1: No

---

## [Editor Report · Acceptance letter]

23 Jul 2021

PONE-D-21-05380R1 

Passive citizen science: the role of social media in wildlife observations 

Dear Dr. Edwards:

I'm pleased to inform you that your manuscript has been deemed suitable for publication in PLOS ONE. Congratulations! Your manuscript is now with our production department. 

Kind regards, 

on behalf of

Dr. Daniel de Paiva Silva 

Academic Editor

PLOS ONE